## RESEARCH ARTICLE

# Super-resolution microscopy reveals a Rab6a-dependent trafficking hub for rhodopsin at the mammalian rod photoreceptor Golgi

Maryam Hekmatara[1], Samantha L. Thompson[1], Kristen N. Haggerty[1], Sydney Hagen[1], Brooke A. Brothers[1], Bali Daniels[2], Guillaume Luxardi[2], Ala Moshiri[2], Wen-Tao Deng[1] and Michael A. Robichaux[1,*]

## ABSTRACT

Rod photoreceptor stability is critical for retinal health and lifelong vision. Rhodopsin (Rho) trafficking is essential for rod homeostasis, as its mislocalization precedes rod cell death in inherited retinal disorders such as retinitis pigmentosa. Despite its importance, the molecular mechanisms of Rho trafficking in mammalian rods remain largely undefined. We investigated Rho's subcellular organization in the mammalian rod Golgi complex. We utilized STORM and structured illumination microscopy super-resolution imaging to map Golgi proteins with Rho in mouse and macaque rods. Our analysis found that a large proportion of Rho in this subcellular region colocalizes with Rab6a in the trans-Golgi. To functionally test this interaction, we utilized a dominant-negative Rab6a mutant in HEK293T cells and mouse rods. The mutant significantly inhibits Rho secretion in cell culture, causing intracellular retention. In mouse rods, the mutant similarly causes significant trans-Golgi Rho retention; however, a majority of Rho protein still escaped the Golgi and reached the outer segment. Together, these findings uncover critical new subcellular details about Rho organization at the Golgi and establish a role for Rab6a as a regulator of Rho protein release from the trans-Golgi in mammalian rods. Our results provide critical insight into the protein trafficking mechanisms essential for long-term photoreceptor health.

KEY WORDS: Rab6a, Opsin, Golgi, Photoreceptors, Retina

## INTRODUCTION

In retinal photoreceptors, the phototransduction machinery is densely packed into membranous discs of the outer segments (OSs), which enables efficient phototransduction and rapid propagation of the visual signal (Wensel et al., 2016). In rods, rhodopsin (Rho), a photosensitive G-protein-coupled membrane receptor, is the most abundant protein in OS discs. Because these discs are constantly renewed throughout the lifetime of rods, Rho and other proteins are

[1]West Virginia University School of Medicine, Department of Ophthalmology and Visual Sciences and Department of Biochemistry and Molecular Medicine, Morgantown, WV 26506, USA. [2]UC Davis School of Medicine, Department of Ophthalmology and Vision Science, School of Medicine, Sacramento, CA 95817, USA.

*Author for correspondence (michael.robichaux@hsc.wvu.edu)

M.H., 0000-0002-7763-658X; A.M., 0000-0002-4277-8094; W.-T.D., 0000-0002-9357-0656; M.A.R., 0000-0002-2559-4912

constantly fluxed through a trafficking pathway that is primarily localized in the rod inner segment (IS). This trafficking flux is critical for rod photoreceptor health, as Rho protein mislocalization is a toxic outcome of both inherited retinal diseases (Athanasiou et al., 2018; Concepcion and Chen, 2010; Potter et al., 2021; Grossman et al., 2011) and retinal detachment (Fariss et al., 1997; Fisher et al., 2005). Nevertheless, the vital subcellular mechanisms that accomplish Rho trafficking remain to be fully determined in mammalian rods.

The Golgi complex plays a vital role in photoreceptor protein trafficking, and it is located in the myoid region of the IS in all vertebrate rod and cone photoreceptors (Cohen, 1960; Mercurio and Holtzman, 1982). After initial core glycosylation of newly synthesized membrane proteins in the endoplasmic reticulum (ER), some glycoproteins are processed in the Golgi to generate mature complex oligosaccharide structures (Murray et al., 2009). In rods, this processing is fundamental for the transport of several integral OS glycoproteins, including Rho, the α1 subunit of the rod cyclic nucleotide-gated (CNG) channel, and guanylyl cyclase-1 (GC-1) (Pearring et al., 2021). Vertebrate Rho protein is specifically glycosylated at highly conserved N-2 and N-15 residues in the N-terminus/extracellular domain (Murray et al., 2009; Fukuda et al., 1979; Liang et al., 1979). Mutations that disrupt the glycosylation sequence of rhodopsin (*RHO*) can cause the autosomal dominant form of retinitis pigmentosa (adRP), a blinding retinal disease characterized by gradual degeneration of rod photoreceptors (Zhu et al., 2004; Sullivan et al., 1993; Murray et al., 2015).

Within the Golgi, protein cargoes are transported sequentially through an endomembrane system, moving from the cis-Golgi through the medial-Golgi before entering the trans-Golgi network (TGN). Throughout these Golgi membranes, N-linked glycans are modified by glycosidase and glycosyltransferase enzymes (Stanley, 2011). The TGN, which extends from the trans-Golgi cisterna, includes post-Golgi transport vesicles that bud from the trans-Golgi for delivery to specific subcellular destinations (De Matteis and Luini, 2008). The trans-Golgi/TGN of photoreceptors is thus a functional sorting station where cargoes are putatively sorted into distinct vesicular carriers destined for the photoreceptor OS, synaptic terminal, or other membrane domains.

Prior research has established a post-Golgi Rho trafficking pathway in amphibian rods (Wang and Deretic, 2014; Deretic et al., 2021). In those cells, Rho protein is enriched in the Golgi, as shown by immunoelectron microscopy (Papermaster et al., 1986, 1985; Bird et al., 1988), immunofluorescence staining (Kandachar et al., 2018; Mazelova et al., 2009; Fresquez et al., 2025), and transgenic Rho-GFP expression (Moritz et al., 2001). This subcellular localization was assigned to the trans-Golgi cisternae based on the budding of post-Golgi carrier vesicles from the TGN, a subdomain also known as 'Golgi exit sites' (Fresquez et al., 2025; Wang et al., 2012).

Critically, the morphology of amphibian rods differs markedly from that of smaller and more compartmentalized mammalian rods, particularly in relation to the IS myoid (Seo and Datta, 2017; Pearring et al., 2013). In amphibian rods, the IS myoid is relatively close to and contiguous with the cell body, while in mammals, it can be up to >50 µm from the cell body and is connected to the cell body by an 'outer fiber' process (Young, 1967; Townes-Anderson et al., 1988). Such species differences have driven research to determine whether Rho trafficking mechanisms are conserved in mammalian species like mice (Ying et al., 2016; Pearring et al., 2017; Haggerty et al., 2024).

In mammalian rods, the precise organization and functional importance of Rho in the Golgi remain elusive. In the mouse retina, immunofluorescence has been used to localize the Golgi to the IS myoid region in many studies (Grossman et al., 2011; Uribe et al., 2016; Keady et al., 2011); however, the organization of Golgi resident proteins has not been thoroughly analyzed. Rho has also been visualized as being enriched in IS myoid with immunofluorescence in mouse retinas (Keady et al., 2011; Kakakhel et al., 2020) and cat retinas (Fariss et al., 1997), and with immunoelectron microscopy in mouse rods (Liu et al., 1997; Meschede et al., 2020; Lewis et al., 2024). Other mouse studies have highlighted the Golgi as a hub of Rho trafficking in the IS myoids, where Rho protein somehow accumulates or is retained (Keady et al., 2011; Lewis et al., 2024; Crouse et al., 2014; Gupta et al., 2025). We previously found that Rho in the IS myoids of mouse rods did not colocalize with the cis-Golgi marker GM130 and may instead be localized to the medial/trans-Golgi (Haggerty et al., 2024).

In the current study, we evaluated the subcellular organization of Rho in the mammalian rod Golgi to better understand this critical stage of the protein trafficking pathway. We hypothesized that the Golgi complex functions as a key trafficking hub where Rho is retained and released by a specific, Golgi-associated factor. To investigate this, we used super-resolution fluorescence microscopy to analyze the organization of the Golgi compartments in the IS myoids of mouse and macaque retinas. We then used a stochastic optical reconstruction (STORM)-based quantitative analysis to map Rho's Golgi accumulation pattern. This analysis revealed that, like in amphibian rods, Rho is specifically colocalized with Rab6a in the trans-Golgi cisterna, prior to its exit into the TGN. Based on this finding, and given that the *in vivo* role of Rab6a in rods has not been previously investigated, we next tested the effect of a dominant-negative Rab6a mutant on Rho trafficking in both cell culture and mouse rods.

## RESULTS

### The subcellular organization of the Golgi complex in mouse rod inner segments

In vertebrate rod and cone photoreceptors, the IS compartment is subdivided into the mitochondrial-rich ellipsoid region and the Golgi-containing myoid region (Fig. 1A). Previous studies used confocal microscopy to map the subcellular organization of the

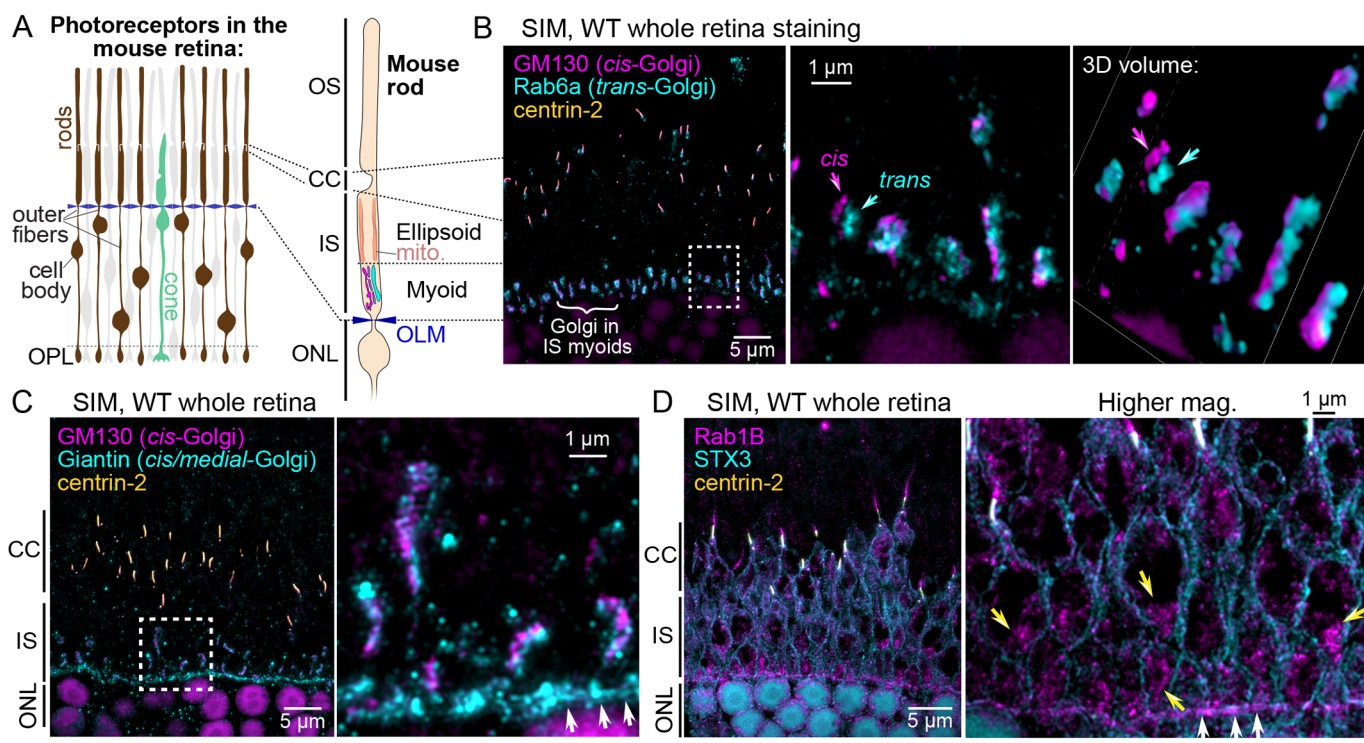

**Fig. 1. SIM localization of Golgi protein markers in mouse rod inner segments.** (A) Diagram depicting the lamination of rod and cone photoreceptors in the mouse retina and the subcellular compartmentalization of mouse rods. The locations of the rod outer fibers, photoreceptor cell bodies, and mitochondria in the IS ellipsoid (mito.) are indicated. The layers/compartments of the mouse rod schematic are aligned with the image in B. (B) Example SIM z-projection image of a WT mouse whole retina immunostained for GM130 (cis-Golgi marker, magenta), Rab6a (trans-Golgi marker, cyan), and centrin-2 (CC marker, yellow). Immunolabeled photoreceptor Golgi are indicated with a white curly bracket. A magnified view of the dashed white line box area is shown, along with a 3D volume view from the same area. In these panels, the labeled Rab6a trans-Golgi can be identified as a distinct Golgi fluorescent puncta (cyan arrows) from the GM130-labeled cis-Golgi (magenta arrows). (C) Example SIM z-projection image of WT whole retina immunostaining for GM130 (magenta), Giantin (cis/medial-Golgi, cyan), and centrin-2 (yellow). In addition to the IS myoids, Giantin is also localized along the outer limiting membrane (OLM, white arrows). (D) Example SIM z-projection image of WT whole retina staining for Rab1B (an ER-Golgi intermediate compartment GTPase, magenta), syntaxin-3 (STX3; a SNARE protein located at the plasma membrane, cyan), and centrin-2 (yellow). Rab1B was localized throughout the IS but was enriched in the IS myoids (yellow arrows) and the outer limiting membrane (white arrows). Throughout, boxed areas represent the high-magnification areas shown to the right. OS, outer segment; CC, connecting cilia; IS, inner segment; ONL, outer nuclear layer; OLM, outer limiting membrane; WT, wild type.

Golgi in the myoids of the large amphibian rod cells (Kandachar et al., 2018; Mazelova et al., 2009; Wang et al., 2012, 2017), which have ISs over 4× wider than mammalian rods (Pearring et al., 2013; Young, 1967; Engbretson and Witkovsky, 1978). To analyze Golgi organization within the relatively thin individual mouse rod ISs, we used whole retina immunofluorescence and structured illumination microscopy (SIM) with validated, Golgi-subdomain-specific antibodies (validation references are in the Materials and Methods). We also included centrin-2 co-immunolabeling to mark the connecting cilia that define the boundary between the rod OSs and ISs (Robichaux et al., 2019).

We first compared the localizations of GM130 and Rab6a, which mark the cis- and trans-Golgi, respectively. GM130, a cis-Golgi structural component and detergent-insoluble matrix protein (Nakamura et al., 1995), is widely used for immunolabeling the position of the Golgi in retinal sections (Grossman et al., 2011; Pearring et al., 2017). The GTPase Rab6a is a master exocytosis organizer enriched in the trans-Golgi cisterna of various eukaryotic cells (Dornan and Simpson, 2023; Echard et al., 1998), including in *Xenopus laevis* rods (Mazelova et al., 2009; Wang et al., 2017). In sections from whole immunostained mouse retinas, both GM130 and Rab6a localized as rows of dense puncta within the photoreceptor IS myoids. High-magnification SIM revealed that, in individual rods, GM130 and Rab6a localized as closely opposed but non-overlapping subdomains, with complete segregation observed in some rods (Fig. 1B). This localization matches the same GM130+Rab6a pattern observed in *X. laevis* tadpole rods, where the cis- and trans-Golgi are reticulated and apposed to each other (Mazelova et al., 2009; Wang et al., 2017). Our findings in mouse rods demonstrate both the specificity of the immunolabeling and the ability to localize the cis- and trans-Golgi using this methodology. The morphology of the immunolabeled Golgi cisternae was variable from rod to rod but was generally elongated and elliptical, consistent with the transmission electron microscopy Golgi ultrastructure (Gupta et al., 2025).

Next, we compared GM130 to Giantin, a large cis-/medial-Golgi matrix protein (Linstedt and Hauri, 1993). In SIM images of mouse rod myoids, GM130 and Giantin partially overlapped (Fig. 1C), demonstrating that Giantin maintains a similar cis-/medial-Golgi localization. Giantin was also prominently localized in a line of fluorescence proximal to the IS myoids, corresponding to the position of the outer limiting membrane (OLM), a junctional boundary between photoreceptor ISs and the outer nuclear layer (ONL) (Fig. 1C, white arrows; Omri et al., 2010). We then mapped the immunolocalization of Rab1B, another GTPase that regulates vesicular trafficking between the ER and cis-Golgi at ER-Golgi intermediate compartments (Plutner et al., 1991). We used syntaxin-3 co-immunolabeling to mark the IS plasma membrane. In SIM images, Rab1B was localized diffusely throughout the rod ISs, including the IS ellipsoids, the connecting cilia (CCs), and the OLM. However, Rab1B appeared partially enriched in the myoid IS (Fig. 1D). This pattern is consistent with the localization of the ER, which extends throughout the IS, including in close proximity to the CC (Thompson et al., 2025), and suggests Rab1B enrichment at an ER-Golgi interface within the myoid.

### Rhodopsin enrichment in rod inner segment myoids

Visualizing Rho protein localization in mouse rod ISs has been technically challenging, requiring additional processing steps and non-traditional tissue preparation (Haggerty et al., 2024). Therefore, to consistently visualize endogenous Rho protein in mouse rod Golgi complexes, we developed a new immunolabeling protocol using vibratome sections of fixed mouse retinas, which are then resin-embedded for thin sectioning. We stained for Rho with the 1D4 monoclonal antibody (hereafter 'Rho-C-1D4'), which targets the Rho C-terminus, i.e. the 1D4 epitope (Molday and MacKenzie, 1983), alongside Rab6a and centrin-2. In these sections, Rho labeling was clear in the ISs, including in a colocalized pattern with Rab6a-labeled trans-Golgi puncta (Fig. 2A,B). This labeling was specific, as Rab6a-labeled cones lacked Rho-C-1D4 staining (Fig. 2B, white arrowheads). Rho was also localized in the ONL, likely in the ER, along the OS tips, and, sporadically, within OS bases. Rho labeling was excluded from the majority of the rod OSs, possibly due to incomplete antibody penetration, consistent with previous whole retina immunolabeling results (Haggerty et al., 2024).

Most thin resin sections we collected for imaging were from the middle of the immunolabeled vibratome sections. In contrast, thin resin sections from the surface of the vibratome sections exhibited much more abundant Rho-C-1D4 immunolabeling. In these surface sections, Rho was still enriched in the IS myoid, but labeling was also evident along the IS plasma membrane and at other internal, cytoplasmic sites (Fig. 2C; Fig. S1). A cross-section through one entire IS was captured from a surface section, providing a complete map of Rho-C-1D4 immunolocalization throughout the rod IS (Fig. S1). This revealed that Rho enrichment in the myoid was more prominent than Rho localization at the IS plasma membrane or the CC membrane.

We also tested alternative immunolabeling and tissue processing methods to confirm that the Rho IS immunolabeling pattern was not just an artifact of our vibratome section labeling protocol. First, we stained deparaffinized thin sections with the Rho N-terminus-targeting 4D2 monoclonal antibody ('Rho-N-4D2'; Hicks and Molday, 1986). Second, we chemically etched unlabeled thin resin sections to uncover antigens for immunofluorescence labeling, based on previously described methodology (Röhlich et al., 1989). Both alternative methods revealed Rho-N-4D2 labeling within the IS myoid and throughout the rod OSs (Fig. 2D,E). However, co-immunolabeling with non-Rho photoreceptor antibodies was not reliable with either of these methods. The deparaffinized sections, in particular, showed evidence of incomplete permeabilization and tissue extraction artifacts. Together, these three staining methods demonstrate that Rho immunolabeling is enriched in mouse rod IS myoids.

### Rhodopsin colocalization with Rab6a in the rod trans-Golgi
### SIM localization imaging

Our previous finding that immunolabeled Rho puncta in rod ISs did not colocalize with GM130 (Haggerty et al., 2024), combined with the Rho-C-1D4+Rab6a colocalization seen in our confocal imaging (Fig. 2A,B), led to our hypothesis that Rho is retained in the rod trans-Golgi. To test this, we used immunolabeled wild-type (WT) mouse retinal vibratome sections for a SIM analysis of Rho and Rab6a colocalization in the IS myoids. In these images, while Rab6a was extensively immunolabeled throughout the mouse rod IS, it was enriched in the rod myoids and colocalized with Rho-C-1D4 immunofluorescence (Fig. 3A). In higher-magnification views of single rod myoids, Rab6a appeared more vesicular and overlapped with Rho, which localized to more continuous, reticulated Golgi-like membranes (Fig. 3A, orange arrows).

To determine if Rho colocalization with Rab6a in the trans-Golgi is conserved in another mammalian species, we performed the same retinal immunolabeling and SIM imaging experiment using adult macaque peripheral retinal tissue. The macaque peripheral retina is rod-dominant, and rod photoreceptors in macaque retinas were shown to be larger than mouse rods (Grünert and Martin, 2020; Packer et al., 1989; Fu and Yau, 2007). To our knowledge, this is the

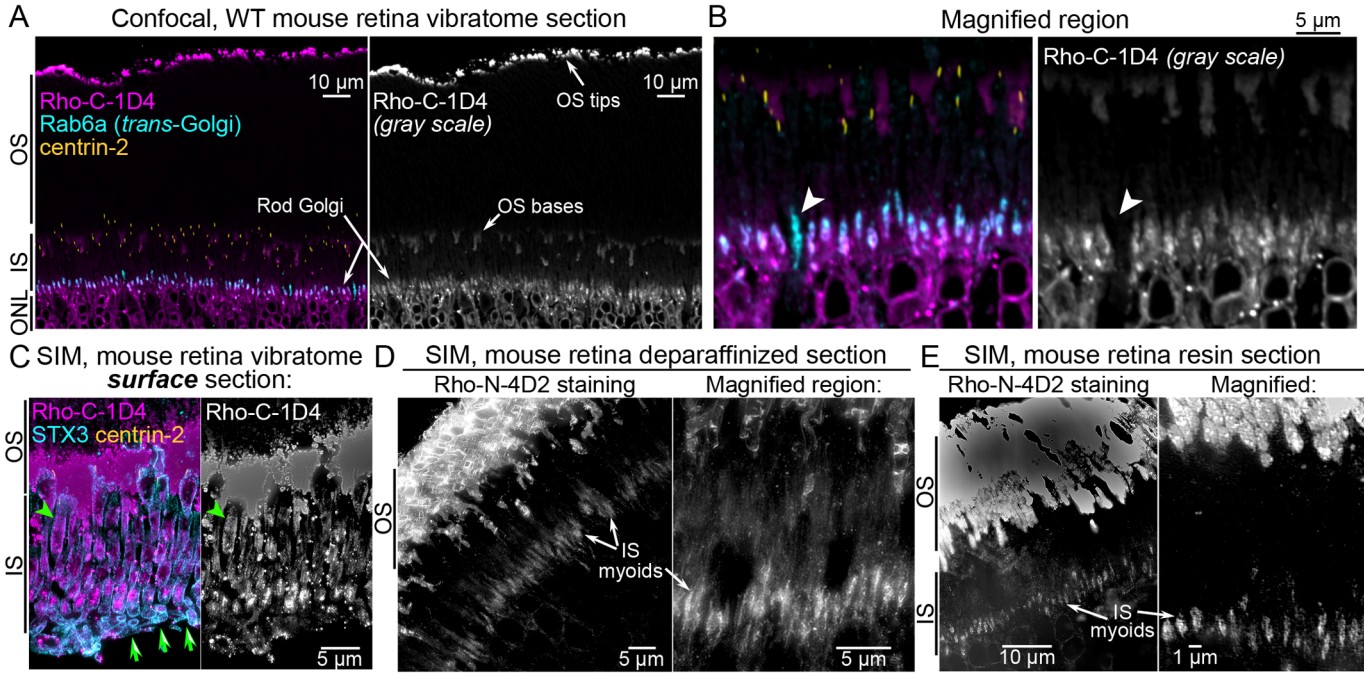

**Fig. 2. Enrichment of rhodopsin immunofluorescence in mouse rod inner segment myoids.** (A) Confocal image of a thin resin section collected from a WT retinal vibratome section that was immunostained for Rho (with the Rho-C-1D4 antibody, magenta), Rab6a (cyan), and centrin-2 (yellow). Rho fluorescent puncta colocalized with Rab6a in the rod IS myoids (indicated as 'Rod Golgi' in the image). Rho-1D4 immunolabeling of the OS tips and bases is indicated; the immunolabeling does not penetrate the middle of the rod OSs. (B) In a magnified region, a Rab6a-labeled cone myoid without any Rho-C-1D4 labeling is indicated (white arrowheads), which demonstrates Rho-C-1D4 labeling specificity. (C) SIM z-projection image of a thin resin section from the surface of a WT retina vibratome surface section immunostained for Rho-C-1D4 (magenta), syntaxin-3 (STX3, cyan), and centrin-2 (yellow). Rho IS staining is more abundant in these rods and localized along the IS plasma membrane (green arrowheads). Green arrows indicate the edge of the vibratome section, demonstrating that this thin resin section was collected from the surface. (D,E) SIM images of Rho-N-4D2 immunolabeling from deparaffinized sections (D) or chemically etched resin sections (E), demonstrating Rho immunofluorescence enrichment in the IS myoids (as indicated). OS, outer segment; IS, inner segment; ONL, outer nuclear layer; WT, wild type.

first immunofluorescence visualization of the Golgi complex in macaque rods. In our SIM images of macaque retinal sections, Rho-C-1D4 was also enriched and highly colocalized with Rab6a, which appeared less vesicular in the trans-Golgi compared to the mouse staining (Fig. 3B).

To compare Rho immunolocalization with the cis-Golgi in mouse retinas, we combined GM130 immunolabeling with the previously validated 'Rho-N-GTX' polyclonal antibody (Haggerty et al., 2024). Unlike Rho-C-1D4, which was not compatible with GM130 multiplex labeling, Rho-N-GTX immunolabeling allowed for co-labeling with GM130. In SIM images of mouse retina vibratome sections, Rho-N-GTX immunofluorescence was detectable in the rod IS myoids but was much less enriched compared to Rho-C-1D4 (Fig. 3C). This difference in labeling may be due to partial antigen masking or differential immunolabeling of the N-terminus in Rho; both are previously reported phenomena for Rho immunolocalization (Röhlich et al., 1989; Vasudevan et al., 2024; Wolfrum and Schmitt, 2000; Besharse et al., 1985). High-magnification views showed that Rho-N-GTX-labeled membranes surrounded but did not overlap with the GM130-positive cis-Golgi. We performed the same immunolabeling and SIM with macaque peripheral retina sections. Again, Rho-N-GTX immunolabeling was less enriched in the IS myoids but was still detectable alongside GM130 (Fig. 3D, white arrows). High-magnification images of macaque Golgi showed that the Rho-N-GTX immunolabeling wrapped around but did not overlap with the cis-Golgi.

In addition to the elongated cis-Golgi in macaque rods, we observed accumulations of shorter, puncta-like GM130 staining in

ISs proximal to the rod Golgi complexes, possibly corresponding to the Golgi in cones of the macaque peripheral retina (Fig. 3D, yellow arrowheads). This dispersed Golgi pattern is consistent with a previous EM ultrastructural observation in peripheral macaque cones (Carter-Dawson and Burroughs, 1992).

**STORM quantitative localization analysis**
Our SIM super-resolution imaging results demonstrated that Rho is specifically retained in the trans-Golgi cisternae of both mouse and macaque rod photoreceptors. To validate these Golgi-related localization observations, we next used STORM single-molecule localization imaging to quantify and statistically compare Golgi marker and Rho immunolabeling patterns from single mouse rods. STORM generates molecular reconstruction maps by capturing photoswitching events from compatible fluorophores and reconstructing them with high-precision Gaussian profiling (Rust et al., 2006). The technique has been utilized to determine nanoscale localization details within various mouse retinal compartments, including rod connecting cilia (Potter et al., 2021; Robichaux et al., 2019; Moye et al., 2025), rod ISs (Haggerty et al., 2024), and ON-type rod bipolar cells (Agosto et al., 2018).

In two-color STORM experiments from immunolabeled mouse retina vibratome sections, we compared Rab6a and Giantin to GM130 in single mouse rod ISs to reveal distinctive molecular distributions (Fig. 4A,B). Similar to our SIM results, GM130 molecules localized as a dense membranous pattern, while both Rab6a and Giantin localized in molecule clusters, suggestive of vesicular labeling. To quantitatively analyze these two-color

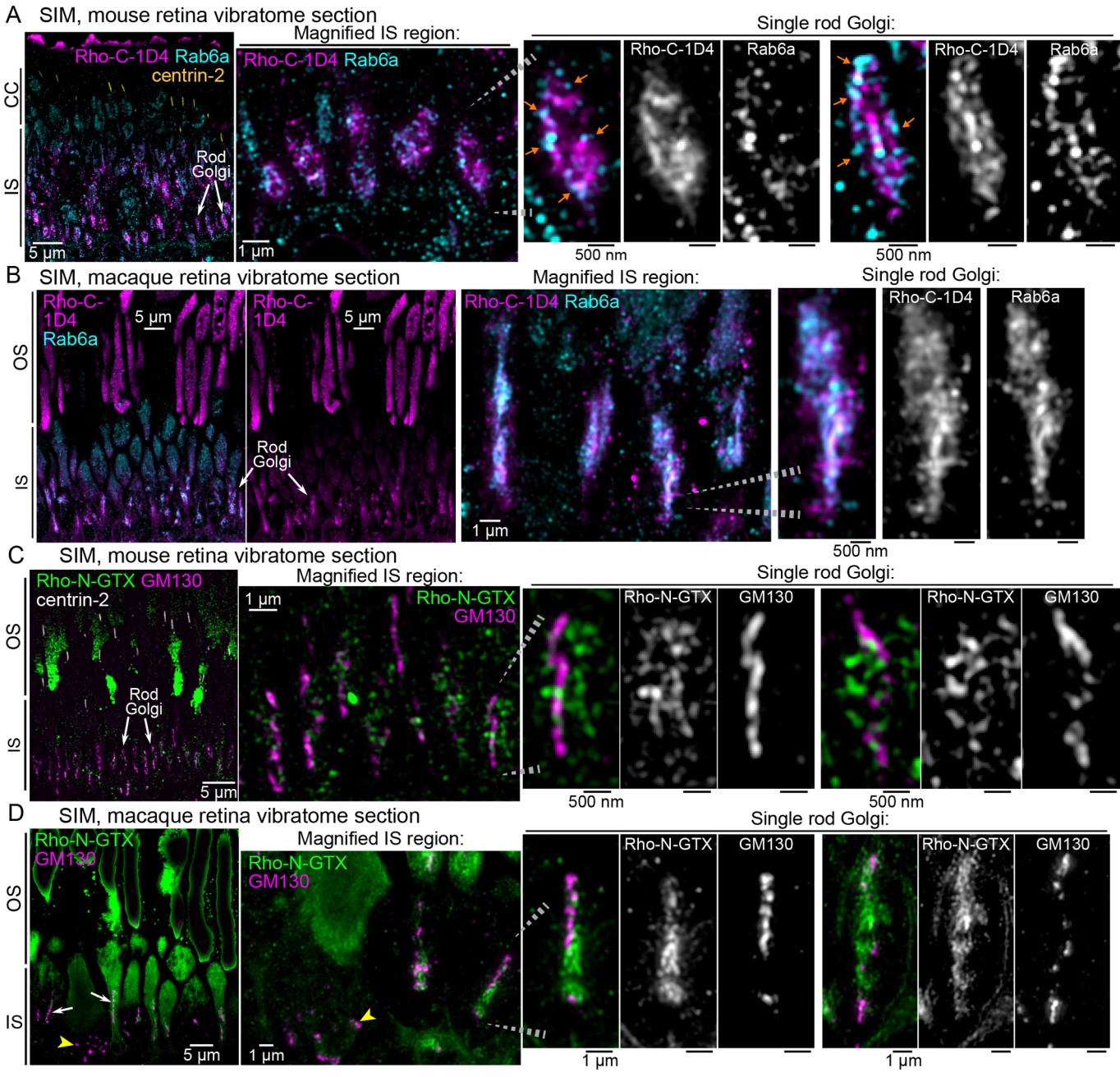

**Fig. 3. SIM analysis of rhodopsin colocalization with Rab6a in the mouse and macaque rods.** (A) A SIM z-projection image of WT mouse retina vibratome sections immunolabeled for Rho-C-1D4 (magenta), Rab6a (cyan), and centrin-2 (yellow). Rho and Rab6a are colocalized in the rod Golgi IS myoids as indicated with white arrows. In the magnified views of single rod Golgi complexes (right panels), vesicular Rab6a puncta colocalized with Rho in a surrounding pattern (orange arrows). The channels are separated from magnified SIM images of rod Golgi and are shown in gray scale throughout the figure. (B) SIM image of macaque peripheral retina vibratome sections immunolabeled for Rho-C-1D4 (magenta) and Rab6a (cyan). Rho and Rab6a also colocalize in macaque rods at the rod trans-Golgi in these IS myoids. Magnified views of the single rod Golgi (right panels) demonstrate robust Rab6a colocalization with Rho. (C) SIM image of WT mouse retina vibratome sections immunostained for Rho-N-GTX (green), GM130 (magenta), and centrin-2 (gray); white arrows indicate the rod cis-Golgi (GM130) at the IS myoid region. In the magnified view of the rod Golgi (right panels), Rho-N-GTX staining is located directly adjacent to but not overlapping with the GM130-stained cis-Golgi. (D) SIM image of a macaque retina vibratome section immunolabeled with Rho-N-GTX (magenta) and GM130 (cyan), showing GM130 and Rho-N-GTX co-labeling of rod Golgi as indicated with white arrows. Yellow arrowheads indicate a unique GM130 staining pattern that could be specific for cones in the peripheral macaque retina. Note some degree of background labeling in this peripheral macaque cone IS, which is attributed to autofluorescence from overfixation. Magnified views of single-rod Golgi complexes demonstrate Rho-N-GTX directly apposed to, but not colocalized with, GM130 in the cis-Golgi. Throughout, the dashed gray lines indicate specific rod Golgi that are magnified in an adjacent panel and scale bar values match adjacent panels when not labeled. OS, outer segment; IS, inner segment; CC, connecting cilia; WT, wild type; SIM, structured illumination microscopy.

STORM maps, we developed a colocalization analysis to measure the overlap between channels. This analysis statistically compared the distances of STORM molecules from a mean position representing the center of each rod Golgi using a two-sample Kolmogorov–Smirnov (K-S) test. The panels in Fig. 4 show examples of the data collected from this analysis: a STORM

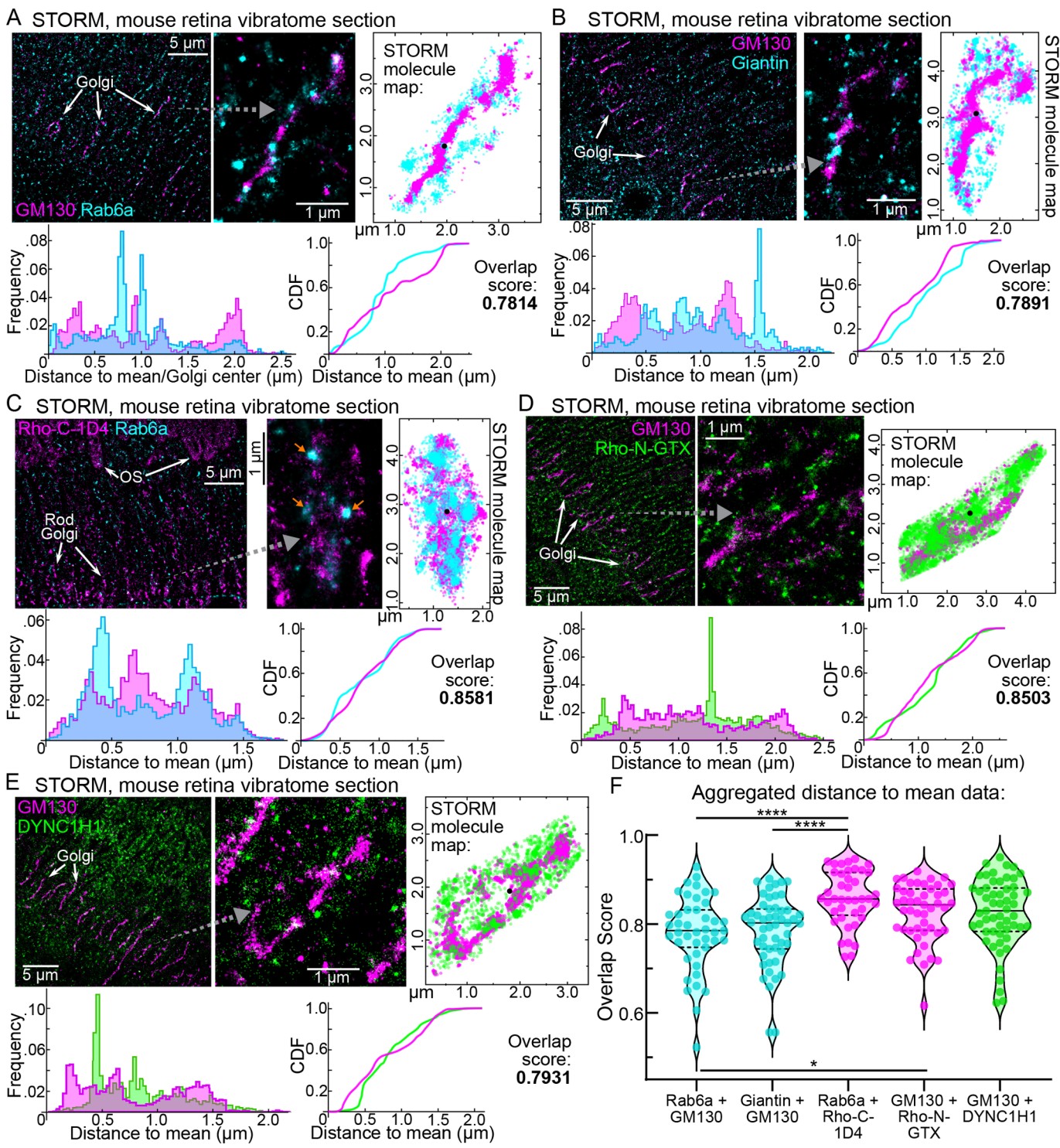

**Fig. 4.** See next page for legend.

molecular map displays the spatial coordinates of the molecules, and frequency distribution and cumulative distribution frequency (CDF) plots show their distances from the Golgi center.

In contrast to the non-overlapping pattern of Rab6a and GM130, STORM reconstructions of Rab6a and Rho (using the Rho-C-1D4 antibody) in mouse IS myoids consistently showed a colocalized pattern (Fig. 4C). In this example, Rab6a molecules were clustered in a vesicular pattern among the overlapping Rho molecules. We also found that Rho-N-GTX molecules were densely localized

throughout the rod IS myoid reconstructions in a partially overlapping pattern with GM130 in STORM maps (Fig. 4D). We included immunolabeling of the dynein-1 complex heavy chain (DYNC1H1) as a diffuse cytoplasmic protein control. Unexpectedly, with both SIM (Fig. S2) and STORM (Fig. 4E), we found some cases where DYNC1H1 partially overlapped with GM130 in mouse rod myoids.

A meta-analysis of the STORM data revealed a statistically significant higher overlap score for the Rab6a+Rho-C-1D4 condition compared to the Rab6a+GM130 and Giantin+GM130

**Fig. 4. STORM single-molecule spatial analysis of Golgi marker proteins and rhodopsin in mouse rod inner segments.** (A) Two-color STORM reconstruction example from a WT mouse retinal vibratome section immunolabeled with GM130 (magenta) and Rab6a (cyan). From a magnified single Golgi example (gray dashed arrow), the GM130 and Rab6a STORM molecule coordinates are plotted (GM130 molecules=6188, Rab6a molecules=3626). Below, distance-to-mean (Golgi center) measurements for GM130 and Rab6a (color-coded as in the images) are plotted in a frequency and a CDF graph. (B-E) STORM reconstructions of retinal sections immunolabeled for Giantin+GM130 (B), Rab6a+Rho-C-1D4 (C), GM130+Rho-N-GTX (D), and GM130+DYNC1H1 (E). Magnified single Golgi are indicated for each with gray dashed arrows. Aligned STORM molecule maps, as well as distance-to-mean frequency and CDF graphs, are provided for each condition as in A. Molecule counts: (B) Giantin=4876, GM130=9291; (C) Rab6a=17,597, Rho-C-1D4=7291; (D) GM130=9896, Rho-N-GTX=13,241; (E) GM130=6009, DYNC1H1=2226. In each example in A-E, the mean position, used to designate the Golgi center, is depicted as a black dot in each STORM molecule map. (F) For each rod Golgi analyzed with STORM, the distance-to-mean distributions were compared with K-S tests, from which we calculated overlap scores, which are compared as a series of violin plots. Overlap scores corresponding to the examples in A-E are provided in each panel. Within the violin plots in F, the median and quartiles are highlighted with dashed lines. The Rab6a+GM130 *n* value (for the number of rod Golgi analyzed from this condition)=43, Giantin+GM130 *n*=51, Rab6a+Rho-C-1D4 *n*=41, GM130+Rho-N-GTX *n*=47, and GM130+DYNC1H1=55. STORM data for each condition were collected from at least two separate WT mouse samples. These aggregated overlap scores for the conditions in our STORM analysis were statistically different based on a Brown-Forsythe ANOVA test (*P*<0.001). Conditions were compared pairwise to test for statistical significance using Dunnett's T3 multiple comparisons tests: Rab6a+GM130 versus Rab6a+Rho-C-1D4 (****P*<0.0001); Giantin+GM130 versus Rab6a+Rho-C-1D4 (****P*<0.0001); Rab6a+GM130 versus GM130+Rho-N-GTX (*P*=0.0495). WT, wild type; STORM, stochastic optical reconstruction microscopy; CDF, cumulative distribution function.

conditions (Fig. 4F). The Rho-N-GTX+GM130 overlap scores were also statistically higher than the Rab6a+GM130 condition, though to a lesser degree. These STORM results statistically demonstrate that a proportion of Rho in the mouse IS myoid region colocalizes with Rab6a in the trans-Golgi.

### Co-immunoprecipitation from mouse retinal lysates
We performed co-immunoprecipitation (co-IP) experiments using detergent extraction from WT mouse retinas to test for an *in vivo* protein-protein interaction between Rho and Rab6a. First, when Rho was captured using anti-1D4 IgG-bound agarose beads, both Rho and Rab6a immunoprecipitated, while the negative control, phosducin (Pdc), did not (Fig. S3A). Second, when Rab6a was specifically immunoprecipitated using protein-A/G beads pre-bound to anti-Rab6a IgG, Rho co-immunoprecipitated (Fig. S3B). Rab6a was not immunoprecipitated from non-specific control IgG-bound beads (Fig. S3C). Together, these co-IP data indicate an interaction – either direct or indirect via a complex with other proteins – between Rab6a and Rho in mouse rods.

### Golgi disruption differentially affects Rho trafficking in cell culture and mouse rods
#### Brefeldin-A Golgi inhibition of mouse explants
We next tested the functional significance of Rho being retained and accumulating in the trans-Golgi of mouse rods, using the pharmacological inhibitor brefeldin-A (BFA), a fungal metabolite that triggers Golgi resorption to the ER (Lippincott-Schwartz et al., 1989). BFA has been shown to disrupt Rho trafficking in cultured cells and in a cell-free assay (Saliba et al., 2002; Deretic and Papermaster, 1991), as well as alter Golgi morphology in *X. laevis*

tadpole rods (Mazelova et al., 2009; Wang et al., 2017). We first validated the effect of BFA on Rho trafficking by treating Rho-GFP-transfected HEK293T cells, a model system where Rho-GFP is normally secreted at the cell surface (Geneva et al., 2017; Chadha et al., 2019). Compared to vehicle-treated control cells, BFA treatment led to intracellular accumulations of Rho-GFP and evidence of Golgi disruption (Fig. S4A,B). We then used WT mouse retinal explants to test BFA's impact on mouse rod Golgi. Post-fixed sections from BFA-treated explants showed mislocalized Rab6a and a significant reduction in GM130 immunofluorescence, with no evidence of BFA-triggered toxicity (Fig. S4C-E). However, despite this clear *ex vivo* Golgi disruption, we found no gross disruption to Rho localization and only minor evidence of redistribution of the trans-Golgi-retained Rho (Fig. S4F,G, yellow arrows).

### Rab6a-T27N dominant-negative disruption of Rho secretion in HEK293T cells
Next, we used the dominant-negative Rab6a-T27N mutant as an approach to disrupt normal Rho release from the trans-Golgi. The Rab6a-T27N mutation forces a GDP-locked, inactive state (Martinez et al., 1994) that has been shown to block cargo export from the trans-Golgi/TGN (Martinez et al., 1994; Matanis et al., 2002; Micaroni et al., 2013). For this study, we generated novel SNAP-Rab6a-T27N or SNAP-Rab6a-WT (control) fusions based on past studies that used stable N-terminal Rab6a fusions (Grigoriev et al., 2007). In transfected HEK293T cells, both fusions localized to the Golgi complex, and the Rab6a-T27N dominant negative mutant caused Golgi expansion in the cytoplasm (Fig. 5A,B).

We next tested the effect of dominant-negative Rab6a-T27N expression on normal Rho trafficking and plasma membrane secretion using a version of Rho-GFP with an additional N-terminal ALFA-tag. In HEK cells transfected with ALFA-Rho-GFP, we observed abundant ALFA-Rho-GFP surface labeling when co-transfected with SNAP-Rab6a-WT (Fig. 5C). In contrast, ALFA-Rho-GFP was depleted from the cell surface and instead accumulated in the Golgi when co-transfected with SNAP-Rab6a-T27N (Fig. 5D). To quantify this surface depletion, we calculated the surface Rho localization ratio by comparing the intensity of ALFA-tag surface labeling with whole-cell GFP fluorescence. This analysis showed that co-expression of SNAP-Rab6a-T27N significantly inhibited ALFA-Rho-GFP secretion (Fig. 5E). Together, these findings indicate that dominant-negative Rab6a-T27N blocks Rho release from the Golgi and subsequent trafficking to the plasma membrane in HEK cells.

### Rab6a-T27N expression in mouse rod photoreceptors with AAV
Based on its significant inhibition of Rho trafficking in HEK cells, we next tested the effect of Rab6a-T27N on Rho trafficking *in vivo*. We generated adeno-associated viruses (AAVs) for both SNAP-Rab6a-WT and SNAP-Rab6a-T27N for subretinal injection into WT mouse retinas. These AAVs contained the minimal mouse opsin promoter for rod-specific expression (Flannery et al., 1997), with a GFP reporter of transduction (Thompson et al., 2025). AAV-transduced retinal sections were analyzed 4 weeks after injection. Both SNAP-Rab6a-WT and SNAP-Rab6a-T27N were predominantly localized in the rod IS myoids (Fig. 6A,B), although we detected low levels of SNAP-Rab6a-WT in transduced rod OSs, potentially as an overexpression artifact. In these transduced regions, GM130 localization was normal in SNAP-Rab6a-WT rods (Fig. 6A), while there were clear instances of mislocalized and redistributed GM130 in rods expressing the dominant-negative SNAP-Rab6a-T27N (Fig. 6B,G). This aberrant GM130 localization indicates that Rab6a-T27N disrupts normal Golgi morphology in mouse rods.

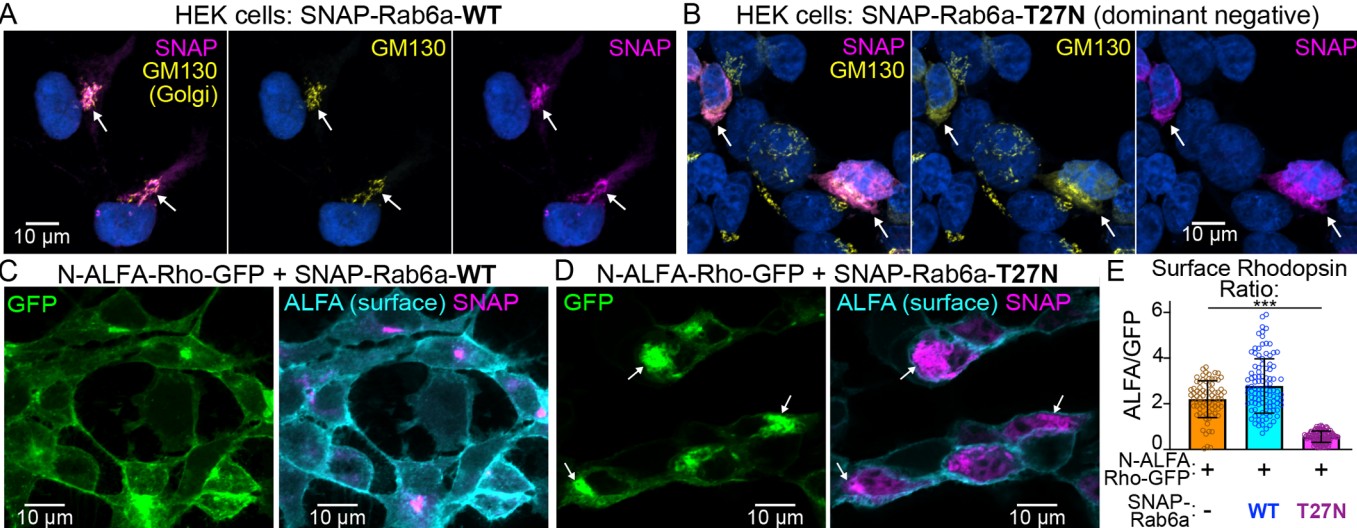

**Fig. 5. Rab6a-T27N dominant-negative expression inhibits rhodopsin plasma membrane trafficking in HEK293T cells.** (A,B) HEK293T cells were transfected with SNAP-Rab6a-WT (A) or SNAP-Rab6a-T27N (B) and labeled with SNAP dye substrate (magenta), anti-GM130 (yellow), and DAPI (blue). White arrows indicate SNAP-Rab6a and GM130 colocalization. (C,D) HEK293T cells co-transfected with N-ALFA-Rho-GFP and either SNAP-Rab6a-WT (C) or dominant-negative SNAP-Rab6a-T27N (D). These cells were surface immunolabeled with ALFA nanobody (cyan) and cell-permeable SNAP dye (magenta). Example confocal images demonstrate that Rab6a-T27N induces ALFA-Rho-GFP aggregation at the trans-Golgi (white arrows). (E) The surface levels of N-ALFA-Rho-GFP were measured based on the ALFA/GFP intensity ratio of a cell. The graph shows that the cells transfected with N-ALFA-GFP and Rab6a-T27N (magenta) had significantly lower N-ALFA-Rho-GFP surface labeling compared to cells with N-ALFA-GFP only (orange) and those co-transfected with SNAP-Rab6a-WT (blue) (***$P$<0.001, one-way ANOVA). Points in the graph=individual cells measured, bars=mean values, error bars=standard deviations. WT, wild-type.

Rho localization in AAV-injected retinas was tested using two methods. First, Rho-C-1D4 immunolabeling of WT retinal vibratome sections was performed. In high-transduction areas with rods expressing SNAP-Rab6a-T27N, there was surprisingly no gross disruption to normal Rho-C-1D4 localization; Rho was still predominantly localized in the OS, similar to SNAP-Rab6a-WT rods (Fig. 6C,D). However, close examination revealed evidence of some Rho-C-1D4 over-accumulation in the IS myoids that colocalized with SNAP-Rab6a-T27N (see inverted and high-magnification panels in Fig. 6D), which indicates a partial blockage of Rho exit from the trans-Golgi. As a second approach, we used WT-RFP knock-in mice, which express human WT-Rho fused to Tag-RFP-T ('Rho-RFP') with an additional C-terminal '1D4' epitope to ensure proper ciliary trafficking (Thompson et al., 2025). AAVs were injected into WT-RFP/+ mice, and Rho-RFP fluorescence was used as a readout of Rho localization. Here again, Rho-RFP was not grossly mislocalized from the OSs in rods transduced with SNAP-Rab6a-T27N (Fig. 6E,F). However, more Rho-RFP appeared to accumulate or become trapped in the IS myoids with SNAP-Rab6a-T27N (see inverted and magnified panels of Fig. 6F).

The localization data from the confocal imaging were quantified by measuring: (1) the fluorescence intensity in the myoid half of the IS compared to the total IS, and (2) total IS fluorescence intensity compared to whole photoreceptor fluorescence. In SNAP-Rab6a-T27N-transduced rods, GM130 immunolabeling was statistically reduced in the myoid compared to SNAP-Rab6a-WT controls (Fig. 6G). This indicates that in some rods (like those in Fig. 6B), the GM130-labeled cis-Golgi was disrupted and redistributed throughout the IS. Furthermore, both Rho-C-1D4 and Rho-RFP were statistically enriched in the IS myoids of SNAP-Rab6a-T27N-transduced rods compared to WT (Fig. 6H,I), demonstrating that SNAP-Rab6a-T27N causes significant retention of Rho in the trans-Golgi. Unlike Rho-C-1D4, Rho-RFP was also significantly mislocalized in the IS relative to total photoreceptor Rho-RFP,

suggesting that the Rho-RFP fusion was more prone to trans-Golgi retention or IS mislocalization due to SNAP-Rab6a-T27N expression. Finally, these measurements showed that SNAP-Rab6a-T27N itself was statistically more enriched in the IS myoid than SNAP-Rab6a-WT (Fig. 6G-I).

Finally, we used SIM to visualize the subcellular localization of Rho with SNAP-Rab6a-WT and SNAP-Rab6a-T27N in mouse rods. Rho-1D4 immunofluorescence was still colocalized with both Rab6a-WT and Rab6a-T27N in the myoids of AAV-transduced rods (Fig. 7A,B). In images from SNAP-Rab6a-T27N retinas, we found evidence of disrupted Rho organization throughout the IS, with the appearance of abnormal vesicles densely labeled for Rho but also containing Rab6a-T27N (Fig. 7B, orange arrowheads). Since these vesicles were located around the periphery of ISs, they could represent accumulations of exocytosed material.

## DISCUSSION

In this study, we mapped the Golgi complex and Rho trafficking in the IS myoid region of mammalian rods on a subcellular scale. We found that the Golgi serves as a key trafficking hub where Rho is retained in the trans-Golgi and specifically colocalizes with Rab6a. In cell culture, expression of a dominant-negative Rab6a-T27N mutant effectively blocked the secretion of Rho to the plasma membrane (Fig. 5). However, in vivo expression in mouse rods had only a minor, albeit significant, effect on Rho trafficking (Fig. 6). Importantly, we found evidence that Rab6a-T27N rod expression also disturbed normal Golgi morphology (Figs 6B,7B). These results indicate that in mammalian rods, while Rho is normally retained in the trans-Golgi of a tightly organized Golgi complex with Rab6a, its delivery to the OS is not completely dependent on this pathway. Collectively, our localization results refine the subcellular map of the Golgi complex in mouse rod photoreceptors (Fig. 8) and provide new insights into how Rho may be released from this region for trafficking to the OS.

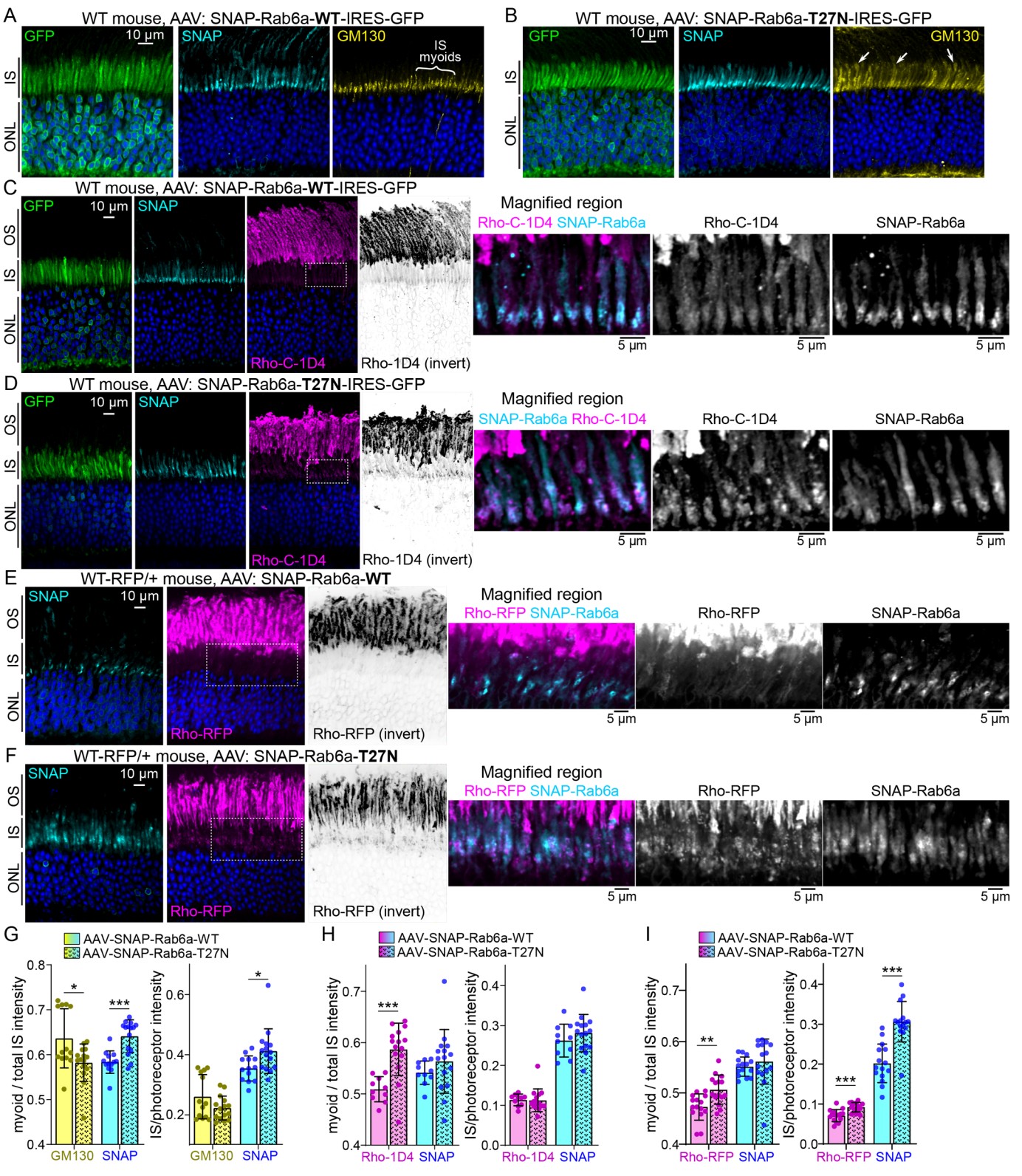

**Fig. 6.** See next page for legend.

Rab6a is a central organizer of Golgi exit, sorting cargoes from the trans-Golgi/TGN toward specific plasma membrane sites in other cell types (Martinez et al., 1994; Grigoriev et al., 2007). A Rab6a-dependent retention and release mechanism from the trans-Golgi is required for the secretion of the transmembrane precursor of tumor necrosis factor (TNF) in macrophages, which was blocked by

the expression of the Rab6a-T27N dominant-negative mutant (Micaroni et al., 2013). Based on these findings and our HEK cell data, we designed our AAV-based approach to express Rab6a-T27N in mouse rods. This genetic manipulation was intended to specifically block Rho exit from the trans-Golgi and bypass the potential compensation from other GTPases. Importantly, while

**Fig. 6. AAV expression of Rab6a-T27N dominant negative slightly disrupts rhodopsin trafficking in mouse rods.** (A-D) Confocal single slice images of retinal sections from WT mice transduced with AAVs for rod-specific expression of SNAP-Rab6a-WT (A,C) and SNAP-Rab6a-T27N (B,D). Both AAVs contain the IRES-GFP transduction marker. In A,B, AAV-transduced retinal sections were immunolabeled for GM130 of the cis-Golgi (yellow) and labeled with DAPI (blue). GM130 was disrupted and mislocalized in the SNAP-Rab6a-T27N-transduced rods in B (right panel, white arrows). (C,D) Rho-C-1D4 (magenta, 'Rho-1D4') immunolabeling was performed for both AAV conditions. (E,F) WT-RFP/+ mouse retinas with AAVs: SNAP-Rab6a-WT (E) and SNAP-Rab6a-T27N (F). Rho-RFP fluorescence is magenta, and sections were co-stained with DAPI to label nuclei (blue). In C-F, the inverted Rho channels highlight the Rho-1D4 or Rho-RFP signal in the IS. Boxed regions are magnified to the far right to highlight Rho localization in the IS myoids. The channels are separated from the magnified region and are shown in gray scale. (G-I) The left graphs depict the ratio of myoid intensities to total IS intensities, while the right graphs depict the ratios of IS intensity to total photoreceptor (OS+IS+ONL) intensities. G contains graphs of GM130 (yellow) and corresponding SNAP-Rab6a (cyan) ratios from AAV-transduced WT retinas, H contains graphs of Rho-1D4 (magenta) and corresponding SNAP-Rab6a (cyan) ratios from WT retinas, and I contains graphs of Rho-RFP (magenta) and corresponding SNAP-Rab6a (cyan) ratios from WT-RFP/+ retinas. In all graphs, data points=transduced retinal regions from confocal single slice images, bars=mean values, error bars=standard deviations. All experiments for each condition were repeated from the retinas of three AAV-injected mice. From retinal sections from each mouse replicate, three to five AAV-transduced retinal regions from single slice images were analyzed and quantified. These results were compiled in the data graphs and compared for statistical significance using unpaired Student's t-tests. *$P<0.05$, **$P<0.01$, ***$P<0.001$. OS, outer segment; IS, inner segment; ONL, outer nuclear layer; WT, wild-type.

Rab6a also functions in retrograde Golgi-ER trafficking, the Rab6a-T27N mutant was shown to have no effect on this pathway (Martinez et al., 1994). Furthermore, based on our imaging in mouse rods, Rab6a appears to assemble at nucleating sites [or 'hotspots' (Miserey-Lenkei et al., 2017)] that may promote the formation of post-Golgi transport vesicles (Figs 3A,4A).

Despite this, we found predominantly normal Rho secretion in Rab6a-T27N-expressing rods. Therefore, assuming that Rab6a-T27N expression blocks endogenous Rab6a Golgi export and that Rho must sequentially pass through the Golgi for release, we propose that an alternative mechanism may facilitate Golgi export of Rho in mouse rods. One alternative could be a novel and unexpected role for the highly homologous Rab6a′ and Rab6b isoforms in mouse rods. However, Rab6a and Rab6a′ have been shown to have non-overlapping roles only in Golgi-to-ER retrograde trafficking (Del Nery et al., 2006), not Golgi export (Micaroni et al., 2013). Rab6b is a neuronal isoform that colocalizes with Rab6a at the Golgi in other cells but is otherwise understudied (Dornan and Simpson, 2023; Opdam et al., 2000). As additional alternative functions, Rab6a may facilitate a Golgi cisternal maturation mechanism, in which Golgi membranes mature progressively from cis-to-trans (Pantazopoulou and Glick, 2019). Rab6a-T27N inhibition of this general process could trigger an atypical, compensatory Golgi exit for Rho and other cargos. Furthermore, membrane proteins have been shown to recycle between the TGN, the plasma membrane, and the endolysosomal system in multiple cell types (Becuwe and Léon, 2014; Lin et al., 2004). Since endocytosis/internalization of the Rho protein has been previously reported (Wang et al., 2014; Ropelewski and Imanishi, 2019), if Rab6a-T27N specifically disrupts a type of cargo recycling in mouse rods, it could account for the minor Rho mislocalization defects that we observed.

Another consideration is the stability and sustained inhibition of AAV-driven Rab6a-T27N in mouse rods. In other GTPases the T27N mutation can result in a nucleotide-free state rather than a GDP-locked state (Macia et al., 2004). This inhibits their function by sequestering guanine exchange factors (GEFs). In such a nucleotide-free state, Rab6a-T27N may be less effective in mouse rods. Finally, Rab6a may facilitate Rho trafficking in a GTPase-independent manner, which could render Rab6a-T27N expression ineffective. Rab6a's C-terminus is geranylgeranylated, which putatively facilitates its membrane insertion (Martinez et al., 1994),

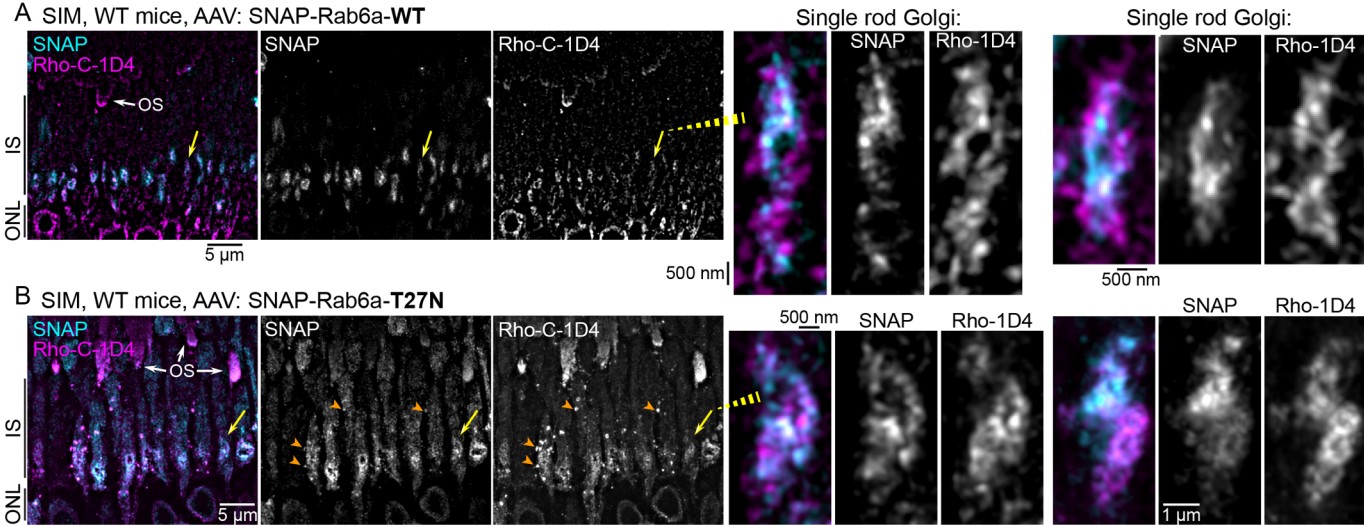

**Fig. 7. Rhodopsin subcellular localization with Rab6a fusions with super-resolution SIM imaging.** (A) Z-projection SIM images of WT mice retina AAV-transduced with SNAP-Rab6a-WT and then immunolabeled for Rho-C-1D4 and co-labeled with SNAP dye substrate. Examples of magnified single-rod Golgi demonstrate the colocalized Rho-1D4 with Rab6a-WT. (B) SIM images of a WT mouse retina AAV-transduced with SNAP-Rab6a-T27N and labeled as in A. These images demonstrate disrupted Rho organization throughout the IS. The orange arrowheads indicate possible exocytosed vesicles that are positive for both Rho-C-1D4 and Rab6a-T27N. Magnified single-rod Golgi images show Rho-C-1D4+Rab6a-T27N colocalization in the trans-Golgi. Yellow arrows indicate magnified single rod Golgi examples. In all magnified SIM images, the single SNAP and Rho-C-1D4 channels are separated and shown in gray scale. OS, outer segment; IS, inner segment; ONL, outer nuclear layer; WT, wild type; SIM, structured illumination microscopy.

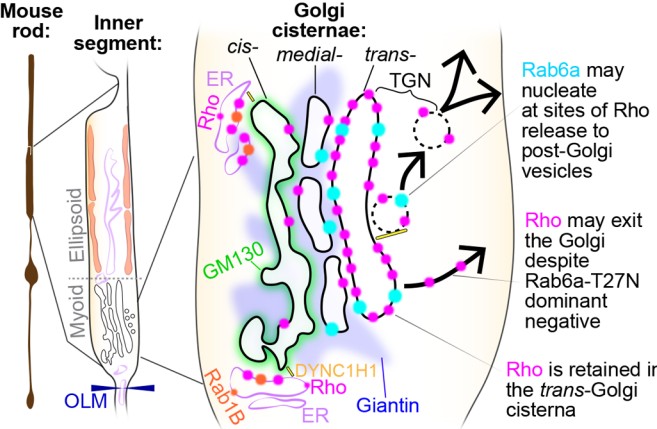

**Fig. 8. Schematic model of rhodopsin (Rho) trafficking through the Golgi complex in mouse rod photoreceptors.** Newly synthesized Rho (magenta) from the ER is transported through the ER-Golgi intermediate compartment, which may contain Rab1B (orange), to the cis-Golgi, which contains GM130 (green). Rho undergoes intra-Golgi transport to the medial (Giantin, blue) and trans-Golgi cisternae. In the trans-Golgi network (TGN), Rho interacts with Rab6a (cyan) for post-Golgi release. DYNC1H1 (yellow) may facilitate Golgi entry and/or exit. OLM, outer limiting membrane; ER, endoplasmic reticulum.

while other studies have demonstrated specific effector protein interactions with Rab6a's C-terminus (Matanis et al., 2002). Nevertheless, the ineffectiveness of disrupting Rab6a falls in a long line of disruptions to proposed trafficking regulators – Arf4, Rab8, Rab11, rootletin, KIF3A, IFT88, and syntaxin 3 – that are overcome by the robust process of Rho OS trafficking in mice (Ying et al., 2016; Pearring et al., 2017; Yang et al., 2005; Jiang et al., 2015; Janecke et al., 2021).

Unlike Rho, some mouse rod N-linked glycoprotein cargoes, like peripherin-2 and ABCA4, at least partially follow an unconventional trafficking pathway that bypasses the Golgi (Pearring et al., 2021; Conley et al., 2019). Similarly, in cultured cells, ciliary polycystin-2 was shown to exit the Golgi early from the cis-Golgi cisterna (Hoffmeister et al., 2011). However, our findings for Rho demonstrate that despite the presence of Rab6a-T27N, Rho still reaches and is retained in the trans-Golgi cisterna. This indicates that this is an obligate step in the sequence of Rho trafficking through the mouse rod IS (Fig. 8). The purpose of this Rho retention remains unclear. It could be a spatio-temporal checkpoint to ensure proper N-glycan maturation, or alternatively, Rho accumulation could be the driver of post-Golgi vesicle budding and extrusion. The latter is supported by previous studies demonstrating that while Rab6a was required for post-Golgi vesicle fission from Golgi 'hotspots', post-Golgi material was still extruded as membrane tubules when Rab6a was inhibited (Miserey-Lenkei et al., 2017, 2010).

Following Rho's release from this Golgi checkpoint, the subsequent post-Golgi pathway(s) that deliver Rho protein to the CC and OS remain unresolved. One potential regulator of this process is IFT20, a component of the ciliary intraflagellar transport (IFT) complex that also localizes to the Golgi in ciliated cells, including mouse rods (Keady et al., 2011; Follit et al., 2006). Tamoxifen-induced deletion of IFT20 mouse rods caused Rho to accumulate in the rod IS myoids (Keady et al., 2011; Crouse et al., 2014), while floxed-IFT20 deletion with iCre75 expression led to Rho mislocalization at the IS plasma membrane and triggered rod degeneration (Lewis et al., 2024). Similarly, deletion of Inpp5e, an IS-localized phosphoinositide phosphatase, caused Rho mislocalization at the Golgi and significant

Golgi morphology defects (Gupta et al., 2025). Intriguingly, the loss of both IFT20 and Inpp5e caused mislocalized Rho to be extruded from the IS as extracellular vesicles (Lewis et al., 2024; Gupta et al., 2025), a phenotype we may have captured in Rab6a-T27N-expressing rods (Fig. 7B). This suggests that extracellular vesicle release may be a common mouse rod response to disruptions to Golgi-localized trafficking events.

As a general cellular mechanism for protein trafficking through the IS, dynein-mediated microtubule transport is essential in mouse photoreceptor health (Dahl et al., 2021). We previously used immunolabeling of the dynein-1 complex heavy chain (DYNC1H1) as a soluble cytoplasmic protein marker in rods (Haggerty et al., 2024). In this study, however, our STORM and SIM imaging revealed that DYNC1H1 labeling was more structured and cytoskeletal, with possible linkages to the rod Golgi (Fig. 4E; Fig. S2). This observation is intriguing, as previous studies have shown that the dynein adaptor proteins, including Bicaudal-D1, Bicaudal-2, and dynactin, interact with Rab6a at the Golgi in cultured cells (Matanis et al., 2002; Short et al., 2002). Meanwhile, both myosin-II and kinesins have been linked with Rab6a and post-Golgi transport in other cell types (Grigoriev et al., 2007; Miserey-Lenkei et al., 2017, 2010). These motor protein complexes would provide mechanical propulsion for both Rab6a-dependent and Rab6a-independent Golgi export of Rho. Future work to identify these mechanisms, alongside continued technical efforts to improve imaging capabilities in mammalian retina models, will be critical for defining the missing components of IS protein trafficking pathway in mammalian photoreceptors.

## MATERIALS AND METHODS
### Animals
All WT mice were C57BL/6J between the ages of 3 weeks and 3 months. WT-hRho-TagRFP-T (WT-RFP) mice were previously described in Thompson et al. (2025) and were in a C57BL/6J background. All mice were housed in a 12 h light/dark cycle. Both sexes were used for experiments in the study unless otherwise noted. All SIM and STORM conditions were repeated from multiple sections from at least two mice. All co-IP experiments were repeated three times using retina samples from three different mice. All experimental procedures using mice were approved by the Institutional Animal Care and Use Committee of West Virginia University (WVU) (approval #2102040326).

Macaque retinal tissue was acquired from the UC Davis primate center. Rhesus macaques were born and sustained at the California National Primate Research Center (CNPRC). The CNPRC is accredited by the Association for Assessment and Accreditation of Laboratory Animal Care (AAALAC) International. Tissue was obtained through the Biospecimen Distribution Program at the CNPRC following approved guidelines performed according to the Institutional Animal Care and Use Committee (IACUC) of UC Davis and the National Institutes of Health (NIH) Guide for the Care and Use of Laboratory Animals, respectively. Tissue samples used for this study were from a 4-year-old male rhesus macaque. Following enucleation, posterior parts were dissected 1 h after death and temporarily stored in Hanks' Balanced Salt Solution (HBSS) for 3 h and then fixed in 4% paraformaldehyde (PFA) for 1 h.

### Antibodies and labeling reagents
The following primary antibodies were used in this study: anti-Rho-C-1D4 (Millipore Sigma, Cat# MAB5356); anti-Rho-N-4D2 (Millipore Sigma, Cat# MABN15); anti-Centrin-2 (Millipore Sigma, Cat# 04-1624); anti-STX3 (Millipore Cat# MAB2258, RRID:AB_1977423); anti-STX3 (Proteintech, Cat# 15556-1-AP, RRID:AB_2198667); anti-Rho-N-GTX (GeneTex, Cat# GTX129910, RRID:AB_2886122); anti-PDC (gift from Dr Maxim Sokolov); anti-GM130 [BD Biosciences, Cat# 610822, RRID: AB_398141, validated by knockout in Han et al. (2017)]; anti-DYNC1H1 (Proteintech, 12345-1-AP); anti-Rab6a [Proteintech, Cat# 10187-2-AP,

RRID:AB_2175463, validated by knockdown in Kinoshita-Kawada et al. (2019)]; anti-Giantin [Proteintech, Cat# 22270-1-AP, RRID: AB_2879055, validated by localization in Ando et al. (2024)]; Rab1B [Proteintech, Cat# 17824, RRID# AB_2237881, validated by knockdown in Sun et al. (2024)]; FluoTag-X2 anti-ALFA-Alexa647 (Nanotag Biotechnologies, Cat# N1502); WGA-CF568 (Biotium, Cat# 29077); and WGA-Alexa-CF488A (Biotium Cat# 29022) were used for surface labeling.

The following secondary antibodies were used in this study: F(ab′)2-goat anti-rat Alexa 488 IgG (Invitrogen, Cat# A11070); F(ab′)2-goat anti-mouse Alexa 488 IgG (Invitrogen, Cat# A11017); F(ab′)2-goat anti-rabbit Alexa 647 IgG (Invitrogen, Cat# A21246); F(ab′)2-goat anti-mouse Alexa 647 IgG (Invitrogen, Cat# A21237); F(ab′)2-goat anti-mouse Alexa 555 IgG (Invitrogen, Cat# A21425); FluoTag®-X2 anti-mouse IgG1 Alexa647 (Nanotag, Cat# N2002-AF647-S, RRID: AB_3076020); IRDye800CW goat anti-rabbit IgG (LI-COR, Cat# 925-32211); IRDye800CW goat anti-mouse IgG (LI-COR, Cat# 925-32210). Other labeling reagents include SNAP-Surface Alexa Fluor 647 [New England Biolabs (NEB), Cat# S9136S], SNAP-Cell® TMR-Star (NEB, Cat# S9105S) and DAPI (Thermo Fisher Scientific, Cat# 62248).

### DNA plasmids
The bovine pcDNA3.1-Rho-EGFP-ETSQVAPA plasmid ('Rho-GFP' in Fig. S4) was obtained from Addgene (#45399). To generate the N-ALFA-Rho-EGFP-1D4 plasmid, we cloned the ALFA tag sequence (Götzke et al., 2019) immediately after the START codon of bovine pcDNA3.1-Rho-EGFP-ETSQVAPA using site-directed mutagenesis (NEB, Cat# E0554S). To generate SNAP-Rab6a-WT and SNAP-Rab6a-T27N plasmids, we replaced the N-terminal myc tag sequences in the PCMV-intron myc Rab6WT (Addgene #46781) and PCMV-intron myc Rab6 T27N (Addgene #46782) plasmids with the SNAPf sequence from the SNAP-Sec61B plasmid (Addgene #5520S) using HiFi DNA Assembly cloning (NEB, #E5520S). All constructs were validated by sequencing, and the new plasmids can be obtained via Addgene.

### Retinal immunofluorescence
#### Whole mouse retina immunolabeling
For whole mouse retina immunolabeling, mice were euthanized and the eyes were enucleated. Retinas were dissected in ice-cold Ames' media (Sigma-Aldrich, Cat# A1420) and immediately transferred to 4% PFA in Ames' media for 5 min on ice. Fixed retinas were then quenched [100 mM glycine diluted in 1× phosphate-buffered saline (PBS)] for 30 min at 4°C and then blocked in SUPER block buffer [15% normal goat serum (NGS; Fitzgerald Cat# 88r-ng001), 5% bovine serum albumin (BSA; Sigma-Aldrich, Cat# B6917)+0.5% BSA-c (Aurion, VWR, Cat# 25557)+2% fish skin gelatin (Sigma-Aldrich, Cat# G7041)+0.05% saponin (Thermo Fisher Scientific, Cat# A1882022)+1× protease inhibitor cocktail (GenDepot, Cat# P3100-005)] for 3 h at 4°C with mild agitation. 2-5 µg of primary antibodies were added directly to the blocking solution for 3 full days of labeling at 4°C with mild agitation. Retinas were washed in 2% NGS diluted in Ames' media six times, for 10 min each wash on ice. Retinas were then probed with fluorescence secondary antibodies, diluted 1:500, at 4°C for 12-16 h with mild agitation. Retinas were washed and post-fixed in 2% PFA for 30 min at 4°C with mild agitation. Retinas were then washed and dehydrated with an ethanol series with the following steps of pure ethanol diluted in water: 50%, 70%, 90%, 100%, and 100%. Resin embedding was performed on dehydrated retinas using Ultra Bed Low Viscosity Epoxy resin (Electron Microscopy Sciences, Cat# 14310) in the following series of steps at room temperature (RT): 1:3 resin to 100% ethanol for 2 h; 1:1 resin to 100% ethanol for 2 h; 3:1 resin to 100% overnight; two steps of full resin (no ethanol), 2 h for each step, with mild agitation for all steps. Embedded retinas were mounted in molds and cured at 65°C for 24 h. 0.5 µm to 1 µm thin retinal sections were sectioned from the resin-embedded retina blocks on a Leica UCT ultramicrotome. Sections were dried onto #1.5 coverslips (Thermo Fisher Scientific, Cat# 12541016) and mounted onto glass slides with ProLong Glass (Invitrogen, Cat# P36980).

#### Mouse retina vibratome section immunolabeling
Corneas were punctured from enucleated mouse eyes in Ames' media and then transferred to 4% PFA for fixing at RT for 15 min with mild agitation.

After removing the cornea, lens, and optic nerve, the posterior segments/eyecups were fixed in 4% PFA for an additional 15 min. For vibratome sectioning, fixed retinas were dissected from the eye cups and embedded in 4% low-melt agarose (Lonza, Cat# 50080). 150 µm vibratome sections were collected on a PELCO EasiSlicer vibratome and stored in 1× PBS. Sections were stained by quenching in 100 mM glycine and blocking with 10% NGS+0.1% Triton X-100 in 1× PBS. 1-2 µg primary antibodies were added directly to the blocking solution, and sections were probed overnight at 4°C with mild agitation. Sections were washed in 1× PBS before incubation with secondary antibodies (diluted 1:500 in 1× PBS) for 2 h at RT. Sections were washed and post-fixed with 1% PFA prior to either mounting onto plain slides with ProLong Glass or ethanol dehydration and resin embedding as described in the section above. Resin-embedded vibratome sections were mounted onto flat resin molds and covered with a piece of ACLAR film (Electron Microscopy Sciences, Cat# 5042510) prior to curing and ultramicrotome sectioning.

#### Macaque retina vibratome section immunolabeling
Small sections (∼2 mm²) of the peripheral retina were dissected from fixed macaque eye posterior segments and then embedded in low-melt agarose for vibratome sectioning and immunolabeling as described in the previous mouse sections. After postfixation, the stained macaque sections were additionally quenched with a 1× TrueBlack solution (Biotium, Cat# 23007) for 1 min and immediately rinsed with 1× PBS. Stained sections were then resin embedded for ultramicrotome sectioning as described in the previous mouse retina vibratome immunolabeling section.

#### Deparaffinized section immunolabeling
Mouse eyes were fixed in Davidson's Fluid (33% formalin, 15% ethanol, 5% acetic acid) for 12-16 h at 4°C with mild agitation and then sent to WVU Electron Microscopy Histopathology and Tissue Bank Core for paraffin embedding and sectioning. 3 µm paraffin sections were collected on coverslips. Sections were deparaffinized by incubating in Neo-Clear (Sigma-Aldrich, Cat# 109843500) for 10 min at RT, followed by incubation in Neo-Clear:pure ethanol (1:1) for 2 min at RT. Sections were then hydrated in the following ethanol steps: 100%, 95%, 75%, 50%, and water; each step was 2 min at RT. Antigen retrieval was performed in boiling 0.01 M citrate buffer (pH 6) for 30 min. Sections were then permeabilized in a permeabilization buffer (0.2% fish skin gelatin+0.25% Triton X-100 in 1× PBS) two times, each for 10 min, and then blocked in permeabilization buffer+5% BSA for 1 h at RT. Sections were then stained in 1 µg of primary antibodies (diluted in permeabilization buffer+1% BSA) for 12-16 h at 4°C. After washing in 1× PBS, sections were reincubated with the permeabilization buffer for 10 min at RT and then stained with fluorescent secondary antibodies (diluted 1:500 in permeabilization buffer+1% BSA) for 2 h at RT. Sections were then quenched with 1× TrueBlack solution for 1 min. The coverslips were then rinsed and mounted onto glass slides with ProLong Glass.

#### Etched resin section immunolabeling
This procedure was adapted from a previously described preparation method (Röhlich et al., 1989). Retinas were dissected from mouse eyes in Ames' media, and then the dissected retinas were immediately fixed in 3% PFA+1% glutaraldehyde. The unstained, fixed retinas were then dehydrated and resin embedded as described in the previous sections. 0.5 µm ultramicrotome retinal sections were dried onto #1.5 coverslips, which were then dipped in a 2.5% sodium methoxide (Sigma-Aldrich, Cat# 156256) (diluted in a 1:1 mix of benzene and methanol) for 1-2 min to etch away the resin surrounding the section. Coverslips were then rinsed in 1:1 benzene:methanol and acetone before drying. Sections were stained with 1 µg of primary antibody (Rho-N-4D2) in 1% BSA diluted in 1× PBS for 12-16 h at 4°C. After washing with 1× PBS, sections were stained with fluorescent secondary antibody (diluted 1:500 in 1% BSA). After more 1× PBS washes, the coverslips were mounted on a glass slide with ProLong Glass.

#### Confocal imaging
Confocal microscopy imaging was performed at RT on a CrestV3 spinning disk system equipped with a Hamamatsu Fusion Gen III sCMOS camera,

and a Plan Apo λD 60×/1.42 NA oil objective was used with Lumencor Celesta 405 nm, 477 nm, 546 nm, and 638 nm excitation lasers. Z-projections were collected from regions of interest using a 0.2-µm Z-section thickness. For all confocal imaging used for fluorescence intensity analyses, the same acquisition settings were applied to both WT and mutant sections.

## SIM
SIM was performed at RT as previously described (Haggerty et al., 2024) using a Nikon N-SIM E microscope system equipped with a Hamamatsu Orca-Flash 4.0 camera and with a SR HP Apochromat TIRF (total internal reflection fluorescence) 100×, NA 1.49 oil immersion objective. Z-projections were collected from regions of interest using a 0.2-µm Z-section thickness. SIM image acquisition and reconstruction were performed on the NIS-Elements Ar software.

## STORM
STORM was performed at RT as previously described (Haggerty et al., 2024) using a Nikon N-STORM 5.0 system. This system is equipped with an Andor iXON Ultra DU-897U ENCCD camera and a SR HP Apochromat TIRF 100×, NA 1.49 oil immersion objective. It features a piezo Z stage and includes 100 mW 405 nm, 488 nm, and 561 nm laser lines, as well as a 125 mW 647 nm laser line. For STORM acquisition, Nikon NIS-Elements software was used. Sections and fields of interest were located using widefield epifluorescence. All STORM was performed in 2D mode with a 40 µm×40 µm field of view. Data were acquired at approximately 33 frames per second, with 40,000 frames collected per channel for each acquisition. For all two-color STORM acquisitions, frames for the 561 nm and 647 nm channels – capturing fluorescent photoswitches from the CF568 and Alexa 647 secondary labels – were collected sequentially. Stringent molecule fitting parameters (Robichaux et al., 2019) were used to fit molecules and exclude any overlapping photoswitching events. The resulting STORM reconstructions were populated as single-molecule events with localization errors less than 20 nm. Chromatic aberration between channels was corrected using an X-Y bead warp calibration, and drift correction was performed using an autocorrelation algorithm.

Prior to STORM acquisition, thin resin sections dried on coverslips were etched in a 1% sodium ethoxide solution at RT for 20 min. After etching, the sections were loaded onto an imaging chamber and immersed in the following STORM imaging buffer: 50 mM Tris (pH 8.0), 10 mM NaCl, 10 mM sodium sulfite, 10% glucose, 40 mM cysteamine hydrochloride (MEA, Chem Impex/VWR, Cat# 102574–806), 143 mM BME, and 1 mM cyclooctatetraene (Sigma-Aldrich, Cat# 138924). Finally, STORM coverslips were sealed with quick-set epoxy resin (Devcon) for imaging.

## Cell culture
HEK293T cells from ATCC were cultured in Dulbecco's modified Eagle's medium (DMEM; Gibco, Cat# 11965092) with 10% fetal bovine serum (FBS; Gibco, Cat# A5670701), 100 units/ml penicillin, and 100 µg/ml streptomycin in a 37°C, 5% CO$_2$ incubator. For transfection, sterile 22×22 mm coverslips were coated with 50 µg/ml poly-D-lysine (Gibco, Cat# A3890401), and cells were seeded at a 1:10 density and incubated for 24 h. They were then transfected with 0.5 µg of DNA plasmid using TransIT-LT1 (Mirus). After transfection, cells were prepared for either live-cell or fixed-cell immunofluorescence. For live-cell surface labeling, cells co-transfected with N-ALFA-Rho-GFP+SNAP-Rab6a-WT/T27N plasmids were washed with warm 1× PBS and probed live with SNAP-Cell TM-STAR (1:500) and FluoTag-X2 anti-ALFA-Alexa647 (1:500) diluted in 5% BSA in DMEM for 1 h at RT. Cells were then rinsed three times with 1× PBS, fixed with 2% PFA for 10 min, and mounted with ProLong Glass for imaging. For fixed immunofluorescence, cells were fixed 24-30 h post-transfection with 2% PFA for 10 min at RT. The cells were quenched with 100 mM glycine for 5 min and incubated in a block buffer (15% NGS+5% BSA+0.5% BSA-c+2% fish skin gelatin+0.1% saponin+1× sodium azide in 1× PBS) for 1 h at RT. Primary antibodies (0.5 µg) and/or SNAP-Surface Alexa Fluor 647 (1:200) (diluted in blocking buffer) were added and incubated overnight at 4°C. Cells were washed with 1× PBS+0.1% Triton X-100 before incubation with secondary antibody (1:500) for 1.5 h at RT.

Finally, cells were washed, counterstained with 0.2 µg/ml DAPI, and post-fixed with 1% PFA for 5 min prior to mounting with ProLong Glass. For BFA (BioLegend, Cat# 420601) experiments, cells were treated with 0.5 µg/ml of BFA diluted in dimethyl sulfoxide (DMSO) for 4 h in a 37°C, 5% CO$_2$ incubator. Cells were then rinsed three times with warm 1× PBS and fixed with 2% PFA for 10 min before post-fixation staining. The cells were not authenticated, nor tested for contamination for this project; however, the cells maintained a healthy morphology and were regularly passaged and assessed for quality.

## Ex vivo culture of mouse retinal explants
Mice were euthanized, and the eyes were enucleated. The cornea, lens, sclera, and optic nerve were all removed in oxygenated Ames' media+HEPES (hereafter Ames') at RT. The retinas (explants) were then immediately transferred into a 35×10 mm dish containing oxygenated Ames' supplemented with either 10 µg/ml BFA diluted in DMSO or just DMSO for control explants. Both control and treated samples were incubated for 2 h at RT and protected from light. Half of the media was replaced with freshly oxygenated Ames' every 30 min. Following incubation, explant retinas were fixed with 2% PFA for 15 min at RT. They were then either processed for vibratome sectioning and staining, as described in the 'Mouse retina vibratome section immunolabeling' section, or immunolabeled and resin-embedded like whole mouse retina samples. To test for toxicity, some explant retina vibratome sections were stained with NucGreen Dead 488 Dye (Invitrogen, Cat# R37109) prior to imaging.

## AAV subretinal injections
The AAV constructs used in this study contain an MOPS500 promoter for mouse rod-specific transgene expression (Flannery et al., 1997). The same open reading frames from the SNAP-Rab6a and SNAP-Rab6a-T27N DNA plasmids were used in the AAV transfer vectors. Additionally, each AAV included an internal ribosome entry site (IRES) linked to a fluorescent tag (EGFP) as a fluorescent marker of transduction in mouse rods. The AAV constructs were designed and purchased from VectorBuilder. AAVs (serotype 2/8) were produced and packaged by the WVU Biochemistry and Molecular Medicine Virology Core. Three adult WT mice were injected with each AAV. Subretinal injections, as previously described (Thompson et al., 2025), were performed as follows. First, mouse eyes were dilated with Tropi-Phen drops (Pine Pharmaceuticals), then mice were anesthetized via intramuscular injection with ketamine (80 mg/kg) and xylazine (10 mg/kg) in sterile 1× PBS. Prior to injection, the AAVs were diluted to 4×10$^{12}$ vector genomes/µl titers with sterile PBS, and 0.1% fluorescein dye was added to the mixture for visualization during the injection procedure. To inject, a 25-gauge needle first punctured the edge of the cornea, then a transcorneal subretinal injection of 1 µl of AAV was performed via inserting a 33-gauge blunt-end needle attached to a 5 µl Hamilton syringe containing the AAV mixture. After injection, Neomycin/polymyxin B Sulfate/Bacitracin Zinc ophthalmic ointment (Bausch & Lomb) was applied to the eyes, and antisedan (Orion Corporation) was intraperitoneally injected to reverse anesthesia. AAV-injected mouse eyes were fixed for vibratome sectioning and immunolabeling as described in the previous section; however, these vibratome sections were first screened for GFP (the transduction marker) to select sections with high transduction and minimal injection damage. Then, depending on the experiment, selected vibratome sections were immunolabeled (as previously described) and either directly mounted onto coverslips with ProLong Glass for confocal imaging or resin embedded to make thin ultramicrotome sections for SIM.

## Co-immunoprecipitation and western blotting
WT mouse retinas were dissected and flash frozen before being lysed via pulse sonication. Two different lysis buffers were used for the different co-IP experiments: (1) 200 µl of T-PER buffer (Thermo Fisher Scientific, Cat# 78510) for anti-Rho IPs, and (2) 200 µl of a lysis buffer containing 20 mM HEPES, pH 7.5, 150 mM NaCl, 5 mM CHAPS, 0.55 mM DTT (VWR, Cat# IC10059701), and 1× protease inhibitor cocktail (GenDepot, Cat# P3100-005) for anti-Rab6a IPs. Lysates were centrifuged, and the collected supernatants were saved as the 'input' fractions. For anti-Rho IPs, input samples were incubated with 20 µl of 1D4 antibody-conjugated

Sepharose beads (∼30% slurry, a gift from Dr Theodore Wensel) with mild agitation for 12-16 h at 4°C. For anti-Rab6a IPs, input samples were first prebound with 16 µl of either anti-Rab6a antibody or non-specific Rabbit IgG Isotype Control antibody (Invitrogen, Cat# 31235) for 12-16 h at 4°C with mild agitation. Following this, 20 µl of Protein A/G magnetic beads (Thermo Fisher Scientific, Cat# 88802) was added for a 2 h incubation at 4°C with mild agitation. After the incubation period, beads were pelleted using either a magnetic stand (for magnetic beads) or by centrifugation (for agarose beads). Supernatants were collected here as the 'unbound' fractions. Beads were then washed four times for 3 min each in 1× PBS+0.1% Tween-20 (PBS-T). Finally, beads were eluted by incubating them with 25 µl urea sample buffer containing 6 M urea and approximately 0.03% Bromophenol Blue diluted in 0.125 M Tris (pH 6.8), supplemented with either 360 mM BME (for anti-Rho IPs) or 1% SDS (for anti-Rab6a IPs).

For western blotting, samples were separated by SDS-PAGE using Novex WedgeWell 10% to 20% Tris-Glycine, 0.1 mm, Mini-Protein Gels (Invitrogen) alongside the Precision Plus Dual Color ladder (Bio-Rad, Cat# 1610374) in Tris-Glycine-SDS running buffer (Bio-Rad, Cat# 1610772). Gels were transferred onto a 0.45 µm pore size Immobilon-FL Transfer PVDF membrane (LI-COR, Cat# 92760001) in Tris-Glycine Transfer Buffer (Bio-Rad, Cat# 1610771)+10% methanol. Membranes were blocked using Intercept Blocking Buffer (LI-COR, Cat# 927-6000) for 1 h while rocking at RT, followed by three washes in PBS-T for 5 min each. Blots were then probed with primary antibodies (1:500 to 1:5000 diluted in PBS-T) with rocking at RT for 1 h. Following the primary labeling, blots were washed and probed with secondary antibodies diluted 1:50,000 in PBS-T for 1 h with rocking at RT. Blots were washed again and imaged for fluorescence on an Amersham Typhoon scanner (GE Healthcare).

### Image analysis and statistics
#### Confocal image analysis
All confocal and SIM image processing and analysis were performed using FIJI/ImageJ (Schindelin et al., 2009). For the GM130 immunolabeling intensity analysis from confocal images of retinal explant sections (Fig. S4E), we measured the integrated densities from the IS layers. The intensity values for each region were calculated by averaging measurements from three slices of z-projection images. To quantify surface rhodopsin-labeling in HEK cells (Fig. 5E), we identified single transfected cells from z-projections. We then calculated the ratio of the ALFA-labeling mean intensity to the GFP mean intensity for each cell, which are the reported 'ALFA/GFP' values. For AAV-transduced retinal sections (Fig. 6), single slices from z-projection confocal images were chosen for analysis. To accurately measure intensities from the myoid half of the IS, the midpoint of the IS layer was computationally defined by using the Straighten tool in FIJI to correct for the retina's curvature. We calculated the ratio of the integrated density from the myoid half of the IS to the total integrated density of the IS, which are the reported 'myoid/total IS intensity' values. Then, we calculated the ratio of the IS integrated density to the combined integrated densities from the OS, IS, and ONL layers, which are the reported 'IS/photoreceptor intensity' values.

### STORM spatial analysis
A spatial analysis was performed on the STORM molecule coordinates corresponding to rod Golgi that showed clear fluorescence widefield pre-STORM images. We used a custom script in Mathematica v13.1.0.0 (Wolfram) for the two-channel spatial analysis of the isolated rod Golgi STORM coordinates. For each single rod Golgi dataset, we plotted the molecule coordinates using the ListPlot function. The mean of the 561 nm channel was calculated using the Mean function and designated as the Golgi center. We then calculated the nearest distance of each molecule to the Golgi center for each channel using the EuclideanDistance function. These distances were plotted as frequency plots or CDF plots using the PDF or CDF functions, respectively, and then exported for statistical comparison.

### Statistical analysis
Unpaired Student's t-tests were performed using GraphPad QuickCalcs (https://www.graphpad.com/quickcalcs/). For STORM data, the distances-to-mean values for each channel from single-rod Golgi were compared using the nonparametric, two-sample K-S test in GraphPad Prism v10.5.0. The

K-S test's D-statistic (D) represents the maximum absolute difference between the CDFs of the two channels being compared. We defined an overlap score for each comparison as 1 - D. These overlap score values were aggregated in Fig. 4F, and then the aggregate values were statistically compared using the Brown-Forsythe ANOVA test with Dunnett's T3 multiple comparisons test. All data throughout this study were visualized as plots using GraphPad Prism software.

### Acknowledgements
The authors thank Dr Abigail Moye for useful discussion during manuscript preparation and Dr Melina Agosto for coding assistance for the STORM analysis. The authors also thank Dr Paolo Fagone and the WVU Department of Biochemistry and Molecular Medicine Viral Core Facility for their assistance with AAV preparation.

### Competing interests
The authors declare no competing or financial interests.

### Author contributions
Conceptualization: M.H., M.A.R.; Funding acquisition: A.M., W.-T.D., M.A.R.; Investigation: M.H., S.L.T., K.N.H., S.H., B.A.B., B.D., G.L., M.A.R.; Methodology: M.H., M.A.R.; Resources: A.M., W.-T.D., M.A.R.; Supervision: A.M., W.-T.D., M.A.R.; Validation: M.H., M.A.R.; Writing – original draft: M.H., M.A.R.; Writing – review & editing: M.H., S.L.T., K.N.H., B.D., G.L., W.-T.D., M.A.R.

### Diversity and inclusion
Proper diversity, equity and inclusion practices were used throughout the study to ensure equitable opportunities for those involved.

### Funding
This work was supported by the National Institute of General Medical Sciences (P20 GM144230 Visual Sciences COBRE grant to WVU), the National Eye Institute (R01 EY030056 to W.D. and R01 EY034123 to A.M.), an unrestricted challenge grant from Research to Prevent Blindness to the WVU Department of Ophthalmology and Visual Sciences, and the support of generous donors (to A.M.). SIM imaging experiments were performed in the WVU Microscope Imaging Facility, which has been supported by National Institutes of Health grants P20GM121322 and P20GM144230, the WVU Cancer Institute, and the WVU Health Sciences Center Office of Research and Graduate Education. The Nikon A1R-SIM is supported by funding from U54GM104942 and P20GM103434. The authors declare no competing financial interests. Open Access funding provided by West Virginia University. Deposited in PMC for immediate release.

### Data and resource availability
All relevant data and details of resources can be found within the article and its supplementary information.

### First Person
This article has an associated First Person interview with the first author of the paper.

### Peer review history
The peer review history is available online at https://journals.biologists.com/bio/lookup/doi/10.1242/bio.062303.reviewer-comments.pdf

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
