## [Peer Review File · Biology Open]

Super-resolution microscopy reveals a Rab6a-dependent trafficking hub for rhodopsin at the mammalian rod photoreceptor Golgi

Maryam Hekmatara, Samantha L. Thompson, Kristen N. Haggerty, Sydney Hagen, Brooke A. Brothers, Bali Daniels, Guillaume Luxardi, Ala Moshiri, Wen-Tao Deng and Michael A. Robichaux

DOI: 10.1242/bio.062303

Editor: Catherine L. Jackson

Review timeline

Original submission:	6 October 2025
Editorial decision:	13 October 2025
First revision received:	27 October 2025
Accepted:	29 October 2025

Original submission

First decision letter

MS ID#: bio.062303

MS Title: Super-resolution microscopy reveals a Rab6a-dependent trafficking hub for rhodopsin at the mammalian rod photoreceptor Golgi

Authors: Maryam Hekmatara, Samantha L. Thompson, Kristen N. Haggerty, Sydney Hagen, Brooke A. Brothers, Bali Daniels, Guillaume Luxardi, Ala Moshiri, Wen-Tao Deng and Michael A. Robichaux

I have now reached a decision on the above manuscript.

The reviewer reports are shown at the bottom of this email or can be accessed, together with a copy of this decision letter, by going to:

As you will see, the reviewers gave favourable reports, but raised some critical points that will require amendments to your manuscript. I hope that you will be able to carry these out, because we would like to be able to accept your paper.

At this stage, we also ask you to ensure your manuscript complies with our formatting guidelines - please see our manuscript preparation guidelines for details. Provided you are able to fully address the referees' comments, we are positive about publication of your paper (we accept over 95% of revision submissions) and therefore hope you won't mind any extra work involved in reformatting your manuscript at this point.

Please upload both a 'clean' version of your Word file, along with a highlighted version clearly showing where you have made changes in the revised manuscript. Please avoid using 'Track changes' in Word files as these are lost in PDF conversion.

I should be grateful if you would also provide a point-by-point response detailing how you have dealt with the points raised by the reviewers in the 'Response to Reviewers' box. Please attend to all of the reviewers' comments. If you do not agree with any of their criticisms or suggestions please explain clearly why this is so.

Reviewer 1

Comments for the author

In this study Hekmatara et al combine several different high-resolution microscopy techniques to localise intracellular rhodopsin to the trans-Golgi in mouse and macaque retinal rod cells. They then demonstrate an interaction between the trans-Golgi localised GTPase Rab6a and rhodopsin, and a partial dependence of rhodopsin on Rab6a for Golgi exit. Overall the manuscript is well-written, the data is very clear, and the authors have shown good attention to detail when reporting on methods and statistical analysis.

I just have a couple of minor suggestions to improve the manuscript:

1) In the abstract the authors write that 'Rho specifically colocalizes with Rab6a in the trans-Golgi', however a lot of Rho in the Golgi is not overlapping with Rab6. This could be re-worded to allow more nuance.

2) In figure 3, separation of the channels in the magnified single rod Golgi images would help the reader interpret the degree of overlap between labels. The reported colocalisation between Rab6a and Rhodopsin is far more convincing in figure 7, where this has been done, than it is here in figure 3. It is also a shame that colocalisation in the myoid is not quantified here from the SIM images, however I concede some analysis of this is done in Figure 4 with STORM data.

3) In figure 4, the increase in overlap score between Rab6a and Rho-C-1D4 over other comparisons is quite small, although significant, yet the authors write that the data 'statistically confirm that Rho specifically and strongly co-localizes with Rab6a'. I would perhaps find an alternative word than 'strongly' to reflect the small degree of enrichment - a considerable amount of rhodopsin does not overlap with Rab6a. Alternatively use of a control comparison between more distant proteins, such as between GM130 and a mitochondrial protein, would give a better indication of the size of change to expect, however this would require more experimental work.

4) Figure 6 would also benefit from showing the individual colour channels for the magnified regions of the retina to better assess overlap.

Reviewer 2

Comments for the author

The authors use high resolution fluorescence microscopy and genetic tools to examine Golgi organization and the trafficking of the photoreceptor protein rhodopsin in mouse rod cells. Overall this study provides compelling evidence that rhodopsin localizes to the trans Golgi and plasma membrane in rod cells. Further, use of a dominant negative Rab6 mutant supports a role for Rab6 in controlling the Golgi export and/or localization of Rho. The data presented are high quality and appropriate controls have been performed. Below I have listed a few minor comments that will improve the manuscript without necessitating further experimentation.

1. The colP experiments do not necessarily support a direct interaction between Rab6 and Rho. The proteins could be coimmunoprecipitated indirectly by pulling down additional factors that are in complex with the targets. In vitro binding assays or additional in vivo experiments are required to support a direct interaction between Rab6 and Rho.

2. I think the manuscript would benefit greatly from additional discussion about alternative models or interpretations of the Rab6 dominant negative experiments. One widely accepted model of Golgi function is cisternal maturation, which is thought to be driven in part by progressive action of Rab GTPases (including Rab6) (Pantazopoulou and Glick 2019). It seems likely that the dominant negative mutant can inhibit the cisternal maturation process rather than specifically affecting the export of select cargoes. In addition, many proteins are known to recycle between the TGN and PM in yeast and mammalian cells (Becuwe and Leon 2014 -elife ; Day et al., 2018 - dev cell). If Rho follows a similar mechanism, defects in localization could be from disrupted endocytosis, export from the TGN, or generally disrupted Golgi function. For example, some papers suggest extensive endolysosomal trafficking of Rho (Wang et al., 2014 - plos biology). A discussion of these alternative possibilities would enhance this manuscript.

Reviewer's Responses to Questions

Experimental quality

Does each figure have the proper controls?

If 'No', please indicate reasons in Comments for Author box below.

Reviewer #1:

- Yes

Reviewer #2:

- Yes

Were the data analyzed using appropriate statistical tests?

If 'No', please indicate reasons in Comments for Author box below.

Reviewer #1:

- Yes

Reviewer #2:

- Yes

Reproducibility

Were experiments performed using adequate number of biological replicates?

If 'No', please indicate reasons in Comments for Author box below.

Reviewer #1:

- Yes

Reviewer #2:

- Yes

Does the methods section provide sufficient detail to permit reproducibility?

If 'No', please indicate reasons in Comments for Author box below.

Reviewer #1:

- Yes

Reviewer #2:

- Yes

Completeness

Are the manuscript's conclusions supported by the data?

If 'No', please indicate reasons in Comments for Author box below.

Reviewer #1:

- Yes

Reviewer #2:

- Yes

Scholarship

Do the authors cite and discuss the merits of data that would argue for and against their conclusion?

If 'No', please indicate reasons in Comments for Author box below.

Reviewer #1:

- Yes

Reviewer #2:

- Yes

Does the manuscript title & abstract accurately reflect the contents of the manuscript, without hyperbole?

If 'No', please indicate reasons in Comments for Author box below.

Reviewer #1:

- Yes

Reviewer #2:

- Yes

First revision

Author response to reviewers' comments

We thank the reviewers for their favorable reports and thorough critique of our manuscript. To address these critical points, we have amended the manuscript into a stronger, revised manuscript. Below is a point-by-point response to all of the reviewers' comments.

Responses are in *blue and italicized*.

Reviewer 1: In this study Hekmatara et al combine several different high-resolution microscopy techniques to localise intracellular rhodopsin to the trans-Golgi in mouse and macaque retinal rod cells. They then demonstrate an interaction between the trans-Golgi localised GTPase Rab6a and rhodopsin, and a partial dependence of rhodopsin on Rab6a for Golgi exit. Overall the manuscript is well-written, the data is very clear, and the authors have shown good attention to detail when reporting on methods and statistical analysis.

We thank the reviewer for this positive assessment of our data and methods.

I just have a couple of minor suggestions to improve the manuscript:

1) In the abstract the authors write that 'Rho specifically colocalizes with Rab6a in the trans-Golgi', however a lot of Rho in the Golgi is not overlapping with Rab6. This could be re-worded to allow more nuance.

We agree with the reviewer on this point and have revised this sentence to state that “Our analysis found that a large proportion of Rho in this subcellular region colocalizes with Rab6a in the trans-Golgi.” [Line 29] For readability, we also slightly adjusted the wording for these Abstract sentences [Lines 29-32]. Of note, we also had to make significant cuts to the original Abstract in order to reach the journal’s 200-word limit for this section of revised manuscript.

2) In figure 3, separation of the channels in the magnified single rod Golgi images would help the reader interpret the degree of overlap between labels. The reported colocalisation between Rab6a and Rhodopsin is far more convincing in figure 7, where this has been done, than it is here in figure 3. It is also a shame that colocalisation in the myoid is not quantified here from the SIM images, however I concede some analysis of this is done in Figure 4 with STORM data.

Based on this comment, we have generated a new version of Figure 3, where the magnified single-rod Golgi images are channel-separated into grayscale images to match the organization of Figure 7. To achieve this, we had to reduce the size of some of the panels. The resulting new figure, we acknowledge, will be more helpful for the reader to assess the overlap of our immunolabeling, and we thank the reviewer for this recommendation. We have also amended the text to the legend for Figure 3.

On the reviewer’s point about quantification of the SIM data: we agree that a reliable quantification of the SIM data would be ideal. However, no quantification methods that we are familiar with match the accuracy of using the same samples to perform STORM (as done in Fig. 4).

3) In figure 4, the increase in overlap score between Rab6a and Rho-C-1D4 over other comparisons is quite small, although significant, yet the authors write that the data 'statistically confirm that Rho specifically and strongly co-localizes with Rab6a'. I would perhaps find an alternative word than 'strongly' to reflect the small degree of enrichment - a considerable amount of rhodopsin does not overlap with Rab6a. Alternatively use of a control comparison between more distant proteins, such as between GM130 and a mitochondrial protein, would give a better indication of the size of change to expect, however this would require more experimental work.

We thank the reviewer for pointing out this overstatement. To address this, we have changed this conclusion statement to “...statistically demonstrate that a proportion of Rho in this subcellular myoid region co-localizes with Rab6a.” [Line 253].

We also appreciate the suggestion that a mitochondrial protein be used in a control condition for our spatial analysis. Unfortunately, we have found that the mitochondria in mouse rods are too distant from the myoid region to be reliably used in analyzing myoid-centric regions of interest. We are still seeking a better control marker for future studies. Currently, though, we lack the right marker that would warrant additional staining experiments to address this point for this manuscript.

4) Figure 6 would also benefit from showing the individual colour channels for the magnified regions of the retina to better assess overlap.

We have also made an updated version of Figure 6 that now includes individual channels for the magnified SIM images. This necessitated reducing the size of the other elements in the figure; however, as with Figure 3, we agree that the new figure will better assist reader interpretation of our SIM data.

Reviewer 2:

The authors use high resolution fluorescence microscopy and genetic tools to examine Golgi organization and the trafficking of the photoreceptor protein rhodopsin in mouse rod cells. Overall this study provides compelling evidence that rhodopsin localizes to the trans Golgi and plasma

membrane in rod cells. Further, use of a dominant negative Rab6 mutant supports a role for Rab6 in controlling the Golgi export and/or localization of Rho. The data presented are high quality and appropriate controls have been performed. Below I have listed a few minor comments that will improve the manuscript without necessitating further experimentation.

We thank the reviewer for their positive feedback about the quality of our data and our experimental design.

1. The colP experiments do not necessarily support a direct interaction between Rab6 and Rho. The proteins could be coimmunoprecipitated indirectly by pulling down additional factors that are in complex with the targets. In vitro binding assays or additional in vivo experiments are required to support a direct interaction between Rab6 and Rho.

We understand this critique of our conclusion related to our colP result. We have amended the conclusion statement that the "... data indicate an interaction—either direct or indirect via a complex with other proteins—between Rab6a and Rho in mouse rods." [Line 263] Additional in vivo experiments would be ideal and could be conclusive; however, optimizing the reagents or mouse lines currently for these experiments would take time and investment beyond the scope of this project. We hope the reviewer understands this limitation and that these experiments are not feasible for this manuscript.

2. I think the manuscript would benefit greatly from additional discussion about alternative models or interpretations of the Rab6 dominant negative experiments. One widely accepted model of Golgi function is cisternal maturation, which is thought to be driven in part by the progressive action of Rab GTPases (including Rab6) (Pantazopoulou and Glick 2019). It seems likely that the dominant negative mutant can inhibit the cisternal maturation process rather than specifically affecting the export of select cargoes. In addition, many proteins are known to recycle between the TGN and PM in yeast and mammalian cells (Becuwe and Leon 2014 -eLife ; Day et al., 2018 -dev cell). If Rho follows a similar mechanism, defects in localization could be from disrupted endocytosis, export from the TGN, or generally disrupted Golgi function. For example, some papers suggest extensive endolysosomal trafficking of Rho (Wang et al., 2014 - plos biology). A discussion of these alternative possibilities would enhance this manuscript.

We thank the reviewer for their expertise and thorough explanation on this point. We have reviewed these articles and used them as the basis for a new section of the Discussion [Line 381]. This new section discusses how our expression of the dominant negative Rab6a could disrupt the cisternal maturation of the Golgi or recycling of material between the TGN, plasma membrane, and endolysosomal system. We believe that the new paragraph adds new context and possibilities for future rhodopsin trafficking research and that it is indeed an enhancement to the manuscript, as the reviewer predicted.

Second decision letter

MS ID#: bio.062303R1

MS Title: Super-resolution microscopy reveals a Rab6a-dependent trafficking hub for rhodopsin at the mammalian rod photoreceptor Golgi

Authors: Maryam Hekmatara, Samantha L. Thompson, Kristen N. Haggerty, Sydney Hagen, Brooke A. Brothers, Bali Daniels, Guillaume Luxardi, Ala Moshiri, Wen-Tao Deng and Michael A. Robichaux

I am happy to tell you that your manuscript has been accepted for publication in Biology Open, pending our standard publication integrity checks. It was accepted on 29th October 2025.

© 2025. Published by The Company of Biologists under the terms of the Creative Commons Attribution License (<https://creativecommons.org/licenses/by/4.0/>).